# The Arabidopsis H3K27me3 demethylase JUMONJI 13 is a temperature and photoperiod dependent flowering repressor

Shuzhi Zheng [1,2], Hongmiao Hu[3,4], Huimin Ren[1], Zhenlin Yang[3,4], Qi Qiu[2,4], Weiwei Qi[1], Xinye Liu[1], Xiaomei Chen[3], Xiekui Cui[2], Sisi Li[5], Bing Zhou [2], Daye Sun[1], Xiaofeng Cao[2,4,6] & Jiamu Du [3,4,7]

In plants, flowering time is controlled by environmental signals such as day-length and temperature, which regulate the floral pathway integrators, including *FLOWERING LOCUS T* (*FT*), by genetic and epigenetic mechanisms. Here, we identify an H3K27me3 demethylase, JUMONJI 13 (JMJ13), which regulates flowering time in Arabidopsis. Structural characterization of the JMJ13 catalytic domain in complex with its substrate peptide reveals that H3K27me3 is specifically recognized through hydrogen bonding and hydrophobic interactions. Under short-day conditions, the *jmj13* mutant flowers early and has increased *FT* expression at high temperatures, but not at low temperatures. In contrast, *jmj13* flowers early in long-day conditions regardless of temperature. Long-day condition and higher temperature induce the expression of *JMJ13* and increase accumulation of JMJ13. Together, our data suggest that the H3K27me3 demethylase JMJ13 acts as a temperature- and photoperiod-dependent flowering repressor.

[1] Ministry of Education Key Laboratory of Molecular and Cellular Biology, Hebei Collaboration Innovation Center for Cell Signaling, Hebei Key Laboratory of Molecular and Cellular Biology, College of Life Sciences, Hebei Normal University, Shijiazhuang 050024, China. [2] State Key Laboratory of Plant Genomics and National Center for Plant Gene Research, Institute of Genetics and Developmental Biology, Chinese Academy of Sciences, Beijing 100101, China. [3] National Key Laboratory of Plant Molecular Genetics, CAS Center for Excellence in Molecular Plant Sciences, Shanghai Center for Plant Stress Biology, Shanghai Institutes for Biological Sciences, Chinese Academy of Sciences, Shanghai 201602, China. [4] University of Chinese Academy of Sciences, Beijing 100049, China. [5] Department of Biology, Southern University of Science and Technology, Shenzhen, Guangdong 518055, China. [6] CAS Center for Excellence in Molecular Plant Sciences, Institute of Genetics and Developmental Biology, Chinese Academy of Sciences, Beijing 100101, China. [7] Institute of Plant and Food Science, Department of Biology, Southern University of Science and Technology, Shenzhen, Guangdong 518055, China. These authors contributed equally: Shuzhi Zheng and Hongmiao Hu. Correspondence and requests for materials should be addressed to D.S. (email: dayesun@mail.hebtu.edu.cn) or to X.C. (email: xfcao@genetics.ac.cn) or to J.D. (email: jmdu@sibs.ac.cn)

Varying day-length (photoperiod) and ambient temperature are two environmental cues that play central roles in plant development, including the regulation of flowering time. As a facultative long-day (LD) plant, *Arabidopsis thaliana* flowers earlier under LD conditions compared to short-day (SD) conditions. Higher ambient temperature also promotes flowering and there is considerable crosstalk between the photoperiodic pathway and the ambient temperature pathway. These environmental cues regulate the expression of key flowering integrators such as *FLOWERING LOCUS T* (*FT*)[1]. In the photoperiodic pathway, *FT* is regulated by the oscillating expression of the transcription factor CONSTANS (CO), which integrates circadian clock and day-length signals[2–6]. In the ambient temperature pathway, the MADS-box domain proteins Flowering Locus M (FLM) and Short Vegetative Phase (SVP) are involved in regulating *FT* expression and thus flowering time[7,8]. SVP protein is degraded and its level declines as temperature increases[8]. FLM responds to ambient temperature changes by switching between protein isoforms FLM-β and FLM-δ[9,10]. FLM-β forms a repressive complex with SVP to prevent flowering, whereas the dominant negative FLM-δ forms an SVP–FLM-δ complex that lacks DNA binding and therefore repressor activity, allowing the activation of flowering[11,12]. The bHLH transcription factor PHYTOCHROME INTERACTING FACTOR4 (PIF4) binds directly to the *FT* promoter in a temperature-dependent manner, and strong binding of PIF4 to *FT* depends on the eviction of H2A.Z nucleosomes induced by high temperature[13,14]. Furthermore, the MYB transcription factor protein Early Flowering MYB Protein (EFM) acts as a convergence point for temperature and light regulation of flowering[15].

Epigenetic regulators modulate chromatin conformation and composition, thus affecting the expression of key flowering integrators[16]. Histone methylation, which has important roles in transcriptional regulation and genome integrity[17], is written by histone methyltransferases and erased by histone demethylases[18]. The highly diverse jumonji domain-containing histone demethylases are classified into subfamilies based on their catalytic domain sequence[19,20]. Demethylases from each subfamily target specific substrates and perform distinct functions. For example, the human Lysine (K)-Specific Demethylase 4 (KDM4), KDM5, and KDM6 subfamilies are H3K9me3/H3K36me3-specific, H3K4me3-specific, and H3K27me3-specific demethylases, respectively[20]. In plants, histone demethylases have plant-specific features and different evolutionary relationships compared with their animal counterparts[19]. For example, plants do not possess the KDM6 subfamily H3K27me3 demethylases[19]. Instead, two known plant H3K27me3 demethylases, EARLY FLOWERING 6 (ELF6)/JUMONJI 11 (JMJ11) and RELATIVE OF EARLY FLOWERING 6 (REF6)/JMJ12, show sequence similarities to the human H3K9me3/H3K36me3 bi-specific KDM4 subfamily demethylases[19,20]. The *elf6* and *ref6* loss-of-function mutants display early and late flowering phenotypes, respectively[21], suggesting that various plant H3K27me3 demethylases influence flowering via different pathways. REF6 genome-wide DNA-binding requires four tandem Cys2His2 zinc fingers and functions to counteract Polycomb-mediated gene silencing[22–24].

Here we investigate the role of histone demethylases in plant flowering through the analysis of Arabidopsis JMJ13, which we show possesses H3K27me3 site-specific demethylase activity in vitro and in vivo. We further determined the crystal structure of JMJ13 in peptide-free and H3K27me3 peptide-bound forms. JMJ13 possesses a unique C4HCHC-type zinc finger, and not the previously predicted C5HC2-type zinc finger, despite the two zinc finger types sharing a similar folding topology. The substrate H3K27me3 peptide is specifically recognized by hydrogen bonding and hydrophobic stacking interactions, providing detailed structural insight into the substrate specificity of a plant H3K27me3 demethylase. In addition, we show that JMJ13 plays a role in temperature- and day-length-regulated flowering. Impaired *JMJ13* function leads to early flowering in both LD and SD conditions at high temperature, but not in SD conditions at low temperature. Our genetic studies suggest that JMJ13 acts as a flowering repressor, which modulates flowering time in a temperature- and photoperiod-dependent manner.

## Results

**JMJ13 specifically demethylates H3K27me3.** We previously identified 21 JmjC domain-containing proteins in the *Arabidopsis thaliana* genome and predicted JMJ13 as one of the 15 potentially active histone demethylases[19]. JMJ13 is a homolog of ELF6/JMJ11 and REF6/JMJ12, the two Arabidopsis KDM4 subfamily H3K27me3 demethylases (Supplementary Table 1)[25,26]. To determine whether JMJ13 is an active demethylase, we performed enzymatic activity assays in vivo using a *Nicotiana benthamiana* leaf-based assay[25,27] (Fig. 1a). In cells where JMJ13-GFP was over-expressed, H3K27me3, but not H3K27me2 and H3K27me1, was markedly reduced (Fig. 1b, c). In contrast, there were no significant differences in the tri-, di- and mono-methylation levels of H3K4, H3K9, or H3K36 sites (Supplementary Fig. 1). The H3K27me3 demethylase activity of JMJ13-GFP was abolished when His293 and Glu295, the two conserved iron-binding amino acids, were replaced by alanine (Fig. 1d, e).

We further performed a MALDI-TOF mass-spectrometry-based in vitro demethylase assay[24]. The recombinant expressed JMJ13 catalytic domain (JMJ13CD, residues 90–578, Fig. 2a) displayed unambiguous demethylation activity against H3K27me3 peptides, but not H3K4me3, H3K9me3, or H3K36me3 peptides, confirming that JMJ13 is an H3K27me3 site-specific histone demethylase (Supplementary Fig. 2a–d). Further assays show that JMJ13 has high H3K27me3 demethylase activity, but no significant activity on H3K27me2 and H3K27me1 (Supplementary Fig. 2d–f). Together, the in vivo and in vitro results demonstrate that JMJ13 is predominately an H3K27me3-specific demethylase.

**Crystal structure of JMJ13CD.** To understand the mechanism of H3K27me3-specific demethylation by JMJ13, we performed structural studies. JMJ13 has a central catalytic domain flanked by flexible regions on both N- and C-termini (Fig. 2a). The crystal structure of the JMJ13 catalytic domain (JMJ13CD), including the predicted jumonji, helical, and zinc finger domains, in complex with the co-factor α-ketoglutarate (α-KG), was determined to 2.4 Å resolution (Fig. 2b and Supplementary Table 2). Despite being in different subfamilies, the overall structure of JMJ13CD resembles the previously reported structures of human KDM5A/B/C and Arabidopsis JMJ14, which are composed of two parts: the jumonji domain and the helical plus zinc finger domains[24,28,29] (Fig. 2b). The jumonji domain adopts a typical α-KG-dependent oxygenase fold with a double-stranded β-helix in the center surrounded by several α-helices (Fig. 2b). A $Ni^{2+}$ ion, which replaced the endogenous $Fe^{2+}$ ion during nickel column purification, and the co-factor α-KG are located in the active center of the jumonji domain (Fig. 2b). The helical domain consists of four long α-helices forming a helical bundle (Fig. 2b).

JMJ13 and other members of the KDM5 subfamily of histone demethylases are predicted to possess a C5HC2-type (5 cysteine residues followed by a histidine and 2 additional cysteine residues) zinc finger domain to coordinate two $Zn^{2+}$ ions. However, our structural analysis showed that the predicted C5HC2 zinc finger of JMJ13 in fact adopts a C4HCHC arrangement, which is composed

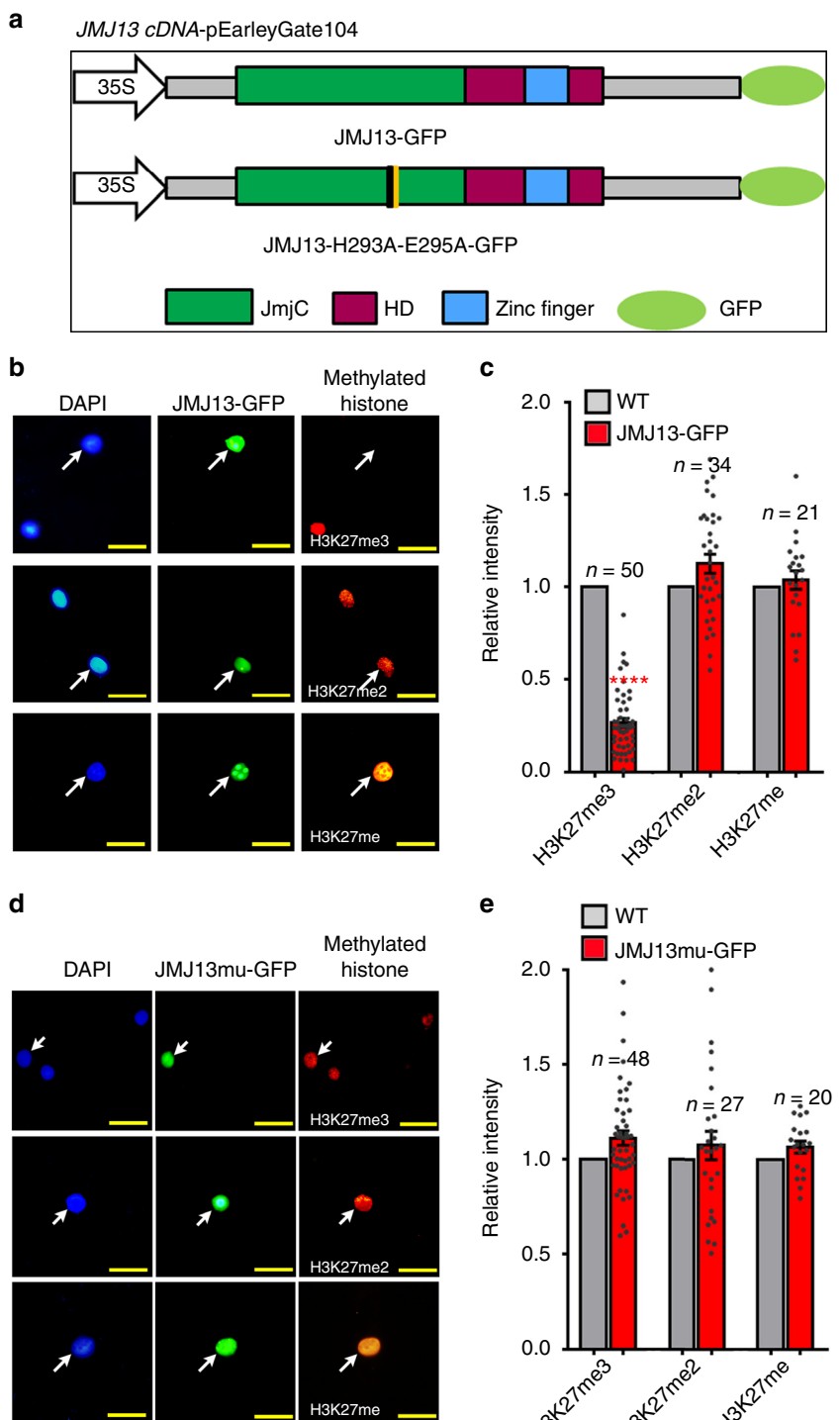

**Fig. 1** JMJ13 is an H3K27me3 demethylase in vivo. **a** Schematic representation of GFP-tagged JMJ13 and JMJ13-H293A-E295A-GFP constructs. HD, helical domain. **b**, **c** Over-expression of JMJ13-GFP reduces the levels of H3K27me3 but not H3K27me2 and H3K27me1 in vivo. **d**, **e** Over-expression of JMJ13-H293A-E295A-GFP has no effect on H3K27 methylation. In **b**, **d**, the white arrows point to the transfected nuclei stained by methylation-specific histone antibodies (red, right panels), DAPI (blue, left panels), and the GFP signal from the JMJ13-GFP or JMJ13-H293A-E295A-GFP (green, middle panels), respectively. Scale bars, 2 μm. In **c**, **e**, more than 20 pairs of transfected nuclei versus non-transfected nuclei in the same field of view were observed and quantifications statistical analyzed. Error bars represent mean ± SE. Student's *t* test was used to calculate the *P* value between JMJ13-GFP and WT. ****$P$ value < 0.0001. The dots denote the individual data points. Source data are provided as a Source Data file.

of a CCCH-type zinc finger (Cys500, Cys503, Cys522, and His525) and a CCHC-type zinc finger (Cys514, Cys516, His519, and Cys534) (Fig. 2b). Rather than the predicted Cys507, the His525

contributes to zinc coordination in the C4HCHC-type zinc finger domain in JMJ13. Although the C4HCHC zinc finger domain is embedded in the primary sequence of the helical domain,

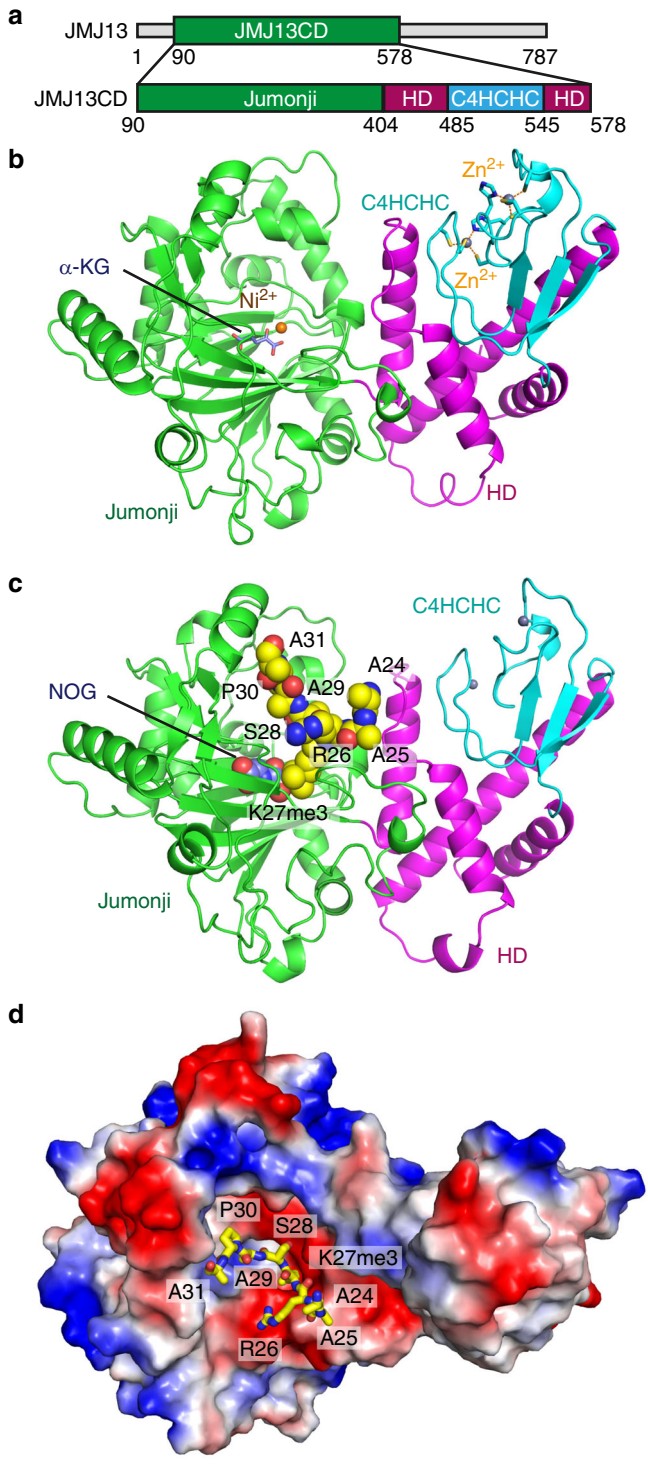

**Fig. 2** Structures of JMJ13-α-KG and JMJ13-NOG-H3K27me3 complexes. **a** A schematic representation of the domain architecture of the JMJ13 (upper panel) and the JMJ13 catalytic domain construct used in crystallization (lower panel). **b** Overall structure of JMJ13CD-α-KG complex in ribbon representation with the jumonji, helical, and zinc finger domains colored in green, magenta and cyan, respectively. The α-KG, nickel ion and zinc ions are shown in stick, orange ball, and silver balls, respectively. The zinc coordination residues are highlighted in stick representation. **c** Overall structure of JMJ13CD-NOG-H3K27me3 peptide complex with JMJ13 in ribbon and NOG and H3K27me3 peptide in space-filling representations, respectively. **d** An electrostatics surface view of JMJ13CD in complex with the H3K27me3 peptide in stick representation showing that the peptide fits into a negatively charge pocket of JMJ13CD. CD catalytic domain, HD helical domain

root-mean-square deviation (RMSD) of 0.49 Å (Supplementary Fig. 3a). The peptide can be traced from H3A24 to H3A31 (Supplementary Fig. 3b). The peptide binds in a negatively charged cleft with the residues H3K27me3 to H3P30 located at the bottom of the cleft and the other flanking residues extending out from the cleft (Fig. 2d). The side chain of H3K27me3 inserts into a deep binding pocket within the active site cleft (Fig. 2d). An NOG molecule and a Ni$^{2+}$ ion are deeply buried in the center of the active site (Fig. 2c).

The interactions between JMJ13 and the H3K27me3 peptide are restricted to the region between H3R26 and H3P30. H3R26 forms salt bridge and hydrogen bonding interactions with Asp236 of JMJ13 (Fig. 3a). H3S28 forms a side-chain hydrogen bond with Asp296 of JMJ13 (Fig. 3a). H3P30 positions its side chain prolyl ring such that the plane of the propyl ring is parallel with and stacks on top of the phenyl ring of Phe179 of JMJ13 (Fig. 3a), resulting in hydrophobic stacking and CH-π interactions. Compared with the JMJ13 peptide-free structure, the side chain of Phe179 undergoes a significant rotation to allow stacking with H3P30, indicative of a peptide binding-induced conformational change (Fig. 3b).

The active site of JMJ13 displays features typical of α-KG-dependent oxygenases observed for other jumonji domain histone demethylases (Fig. 3c)[20,30]. The trimethyllysine docks into the deep binding pocket with the three methyl groups anchored by an extensive CH–O hydrogen-bonding network (Fig. 3c), which is essential for fixing the conformation of the head group of the bound trimethyllysine. A Ni$^{2+}$ ion and an NOG molecule are coordinated by surrounding residues (Fig. 3c).

To dissect the catalytic mechanism, we performed a structure-based mutagenesis study. To that end, we produced the JMJ13$^{D236A}$ and JMJ13$^{D296A}$ proteins, which have alanine replacements of the key residues involved in recognition of H3R26 and H3S28, respectively. These proteins showed significantly reduced demethylase activity (Fig. 3d). Mutation of Phe179 to serine, a small hydrophilic residue, produced a moderate decrease of activity, and replacement by glutamine, a hydrophilic residue with a larger side chain, significantly impaired activity (Fig. 3d), confirming the importance of the hydrophobic interaction between Phe179 and H3P30. Mutations of trimethyllysine or the nickel ion-binding residues completely abolished the activity of JMJ13, revealing their essential role in catalysis (Fig. 3d).

The H3K27me3 mark resides within the same ARKme3S consensus motif as the H3K9me3 mark, making the two marks chemically difficult to distinguish. In the JMJ13 structure, the specific stacking interaction between H3P30, at the $n + 3$ position, and Phe179 selects against the H3K9me3 substrate, which has an H3G12 at the $n + 3$ position, thereby ensuring substrate specificity.

structurally it forms an independent domain that interacts with the helical domain (Fig. 2a–b) in a similar manner to the helical and C5HC2 domains of JMJ14[24].

**Recognition of the H3K27me3 peptide by JMJ13.** To investigate the substrate recognition and catalytic mechanism of JMJ13, we determined the crystal structure of JMJ13CD in complex with the α-KG analog N-oxalylglycine (NOG) and an H3K27me3 peptide, at 2.6 Å resolution (Fig. 2c and Supplementary Table 2). Overall the structure closely resembles the JMJ13-α-KG complex with a

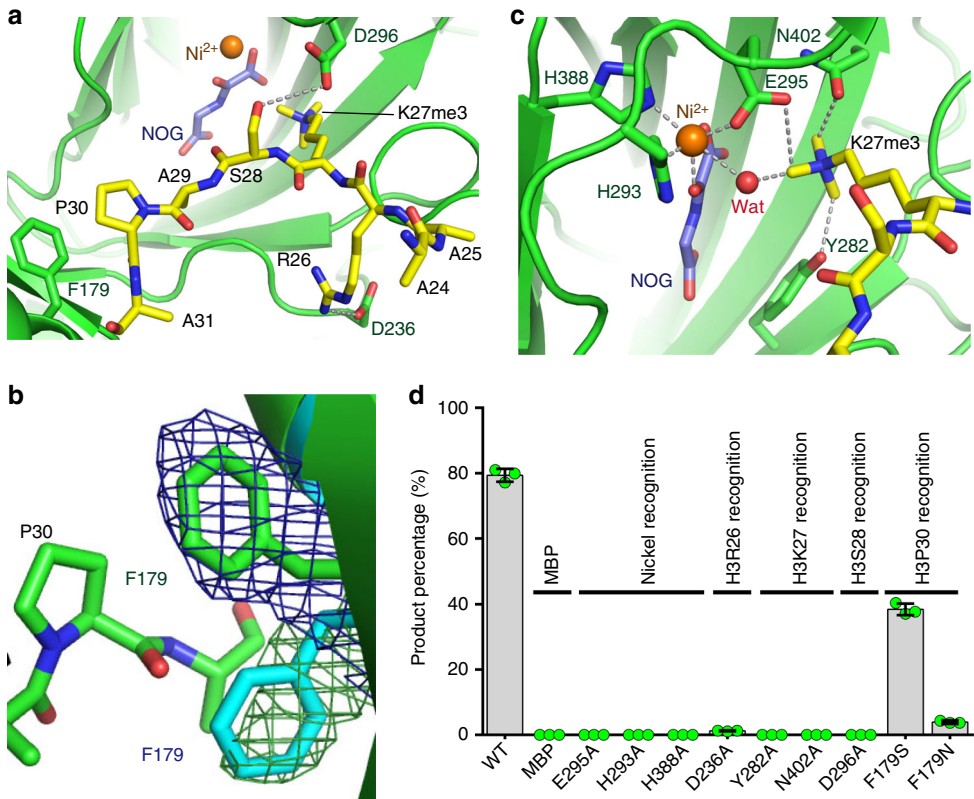

**Fig. 3** Structural basis for the recognition of H3K27me3 by JMJ13. **a** H3R26 and H3S28 form hydrogen bonds (dashed silver lines) with JMJ13 Asp236 and Asp296, respectively. The prolyl ring of H3P30 stacks with the phenyl ring of JMJ13 Phe179. **b** The superposition of JMJ13-α-KG complex (in cyan) and JMJ13-NOG-H3K27me3 complex (in green) shows that the binding of H3K27me3 peptide induces a significantly conformational change of the side chain of Phe179. The SIGMAA weighted 2Fo-Fc maps at 1 sigma level of the Phe179 in the two complexes are shown in meshes. **c** The methyl groups are specifically anchored by surrounding CH–O hydrogen bonds. The nickel ion is coordinated by NOG, a water molecule, and surrounding residues. **d** In vitro H3K27me3 demethylation activity assay of MBP-tagged JMJ13CD and its mutants showing the mutations of key residues involved in peptide binding and catalysis are decreasing the activity of JMJ13. The MBP protein was used as a negative control. The percentages of the product peptide are shown as means ± SD ($n = 3$). Green dots denote the individual data points. Source data are provided as a Source Data file

**Structural comparison with other histone demethylases.** JMJ13 belongs to the KDM4 subfamily of histone demethylases based on phylogenetic sequence analysis of the catalytic domain[19]. However, JMJ13 possesses a domain architecture that does not resemble the two known KDM4 subfamily H3K27me3 demethylases, REF6 and ELF6[19], which contain four tandem DNA-binding C2H2 zinc fingers at the end of the C-terminus (Supplementary Table 1). Instead, JMJ13 resembles the H3K4me3-specific KDM5 subfamily histone demethylases, such as human KDM5/JARID1 and Arabidopsis JMJ14[24,28,29], which contain a fused helical-zinc finger cassette to the C-terminus of the jumonji domain (Supplementary Table 1). In human, KDM4 subfamily demethylases are H3K9me3/H3K36me3 bi-specific enzymes; moreover, all known human H3K27me3 demethylases belong to the KDM6 subfamily, which is absent in plants (Supplementary Table 1)[19,20]. Thus, it is intriguing that a plant KDM4 subfamily demethylase employs a KDM5 subfamily-like domain architecture to conduct the functions of human KDM6 subfamily demethylases.

We superposed our JMJ13–H3K27me3 complex and the human UTX–H3K27me3 complex (PDB code: 3AVR) based on the jumonji domain (Fig. 4a)[31,32]. The jumonji plus helical domains of the two structures are similar (Fig. 4a). The region in UTX corresponding to the JMJ13 C4HCHC domain is the zinc binding domain, which coordinates only one zinc ion (Fig. 4b)[31]. The zinc finger domains of JMJ13 and UTX are of similar topology (Fig. 4b)[31]. The catalytic centers of the two structures

share very similar conformations (Fig. 4a)[31]. The two substrate peptides from the two complexes possess the same directionality and have similar conformations (Fig. 4a and c). Asp1089 of UTX occupies the equivalent position of Asp236 in JMJ13, which is involved in H3R26 recognition, indicating a similar H3R26 recognition mechanism (Fig. 4c)[31]. H3S28 is recognized by Asp296 in JMJ13, but UTX has no specific interaction with H3S28 (Fig. 4c)[31]. H3P30 stacks with Phe179 in JMJ13, but with Pro1144 in UTX, which is in a different and non-homologous position (Fig. 4d)[31]. The recognition of H3P30 is essential for distinguishing between H3K27me3 and H3K9me3, indicating that JMJ13 and UTX have independently evolved different specific stacking interactions with H3P30 to ensure substrate specificity[31].

The KDM4 subfamily member JMJ13 possesses a jumonji-helical-zinc finger domain-like arrangement of the catalytic fragment, like the KDM5 subfamily member JMJ14 (Supplementary Table 1)[24]. The two proteins have similar domain arrangements and overall structures with an RMSD of 1.99 Å (Fig. 4e)[24]. Besides the conserved jumonji and helical domains, the peptide backbone of the C4HCHC zinc finger of JMJ13, and the C5HC2 zinc finger of JMJ14 are quite similar (Fig. 4f)[24]. However, the zinc coordination topologies differ. The first $Zn^{2+}$ ion occupies an analogous position in both JMJ13 and JMJ14. However, the second zinc ion occupies different sites in the two proteins and each is coordinated differently (Fig. 4f–h), yet the proteins maintain very similar

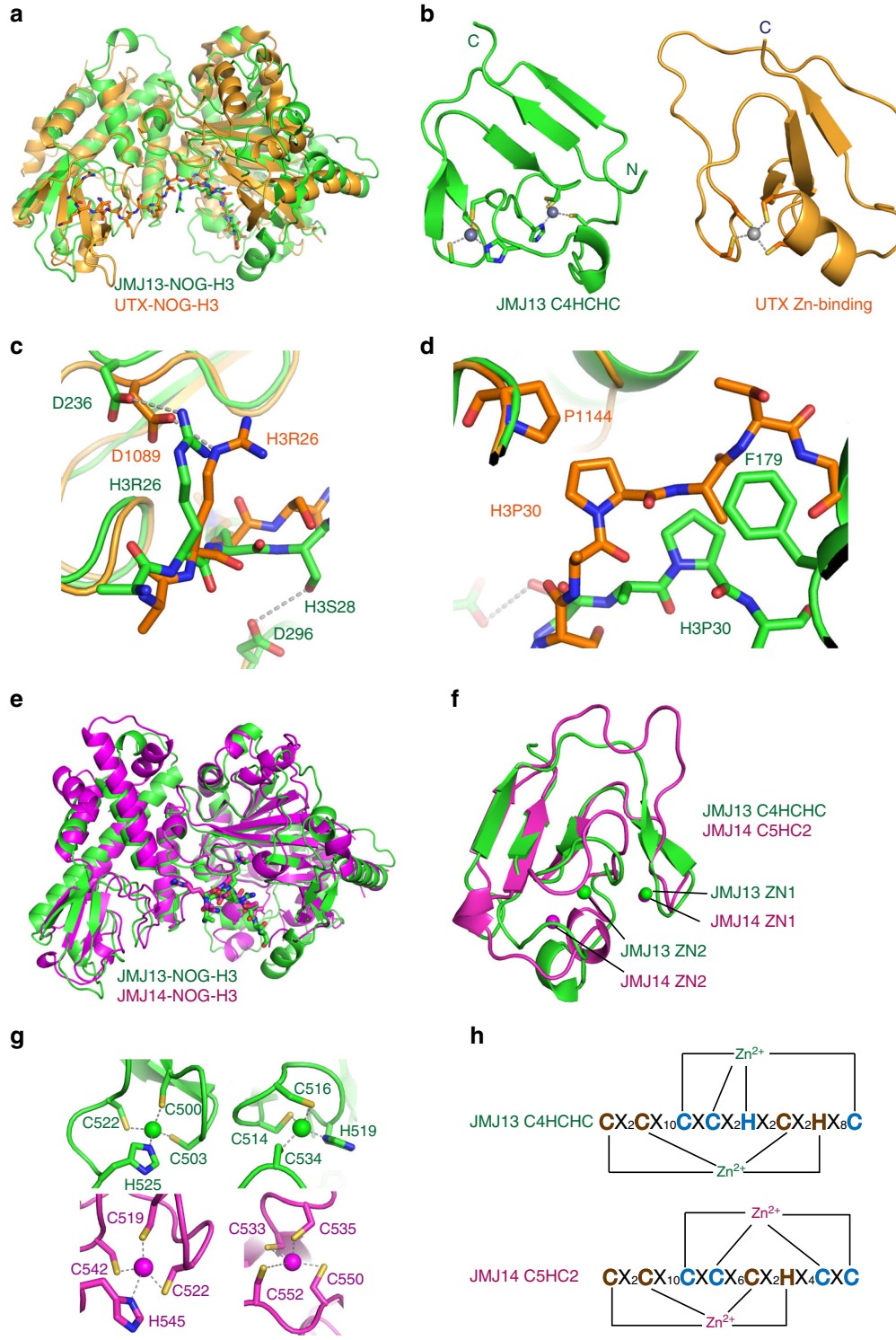

overall zinc finger domain topologies (Fig. 4f–h). Whether this relationship is indicative of similar function is presently not known.

Our structure-based sequence alignment identified the key residues involved in H3K4me3 recognition by JMJ14 and H3K27me3 recognition by JMJ13. These key residues are not conserved (Supplementary Fig. 4), although generally the JMJ13 and JMJ14 show similar overall structures (Fig. 4e). It suggests that the conserved jumonji-helical-zinc finger cassette may function as a general scaffold for histone demethylases, with

specific features evolving within individual demethylases to enable differential regulation and substrate specificity.

**Ectopic *JMJ13* expression causes pleiotropic defects**. H3K27me3 functions as a repressive histone mark[18,33]. To investigate the function of JMJ13, which removes H3K27me3, we generated JMJ13-GFP (*JMJ13ox*) over-expressing transgenic plants. *JMJ13ox* plants had pleiotropic phenotypes, with dwarf seedlings, early flowering, and upward curling leaves

**Fig. 4** Comparison of JMJ13 and other histone demethylases. **a** A superposition of the JMJ13-NOG-H3K27me3 complex (in green) and UTX-NOG-H3K27me3 (in orange, PDB code: 3AVR) shows the similar overall structures and peptide-binding sites. **b** The structures of the C4HCHC dual zinc finger of JMJ13 (left panel, in green) and the C4 single zinc finger of UTX (right panel, in orange) showing different zinc coordination but similar folding topology. **c** The superposition of the JMJ13-NOG-H3K27me3 complex (in green) and the UTX-NOG-H3K27me3 (in orange) shows that the two proteins employ correspondent residues Asp236 of JMJ13 and Asp1089 of UTX to recognize H3R26 in the two complexes, respectively. In contrast, JMJ13 recognize H3S28 by a side chain hydrogen bond with Asp296, but UTX lacks this recognition. **d** The superposition of the JMJ13-NOG-H3K27me3 complex (in green) and UTX-NOG-H3K27me3 (in orange) shows that JMJ13 recognizes H3P30 by Phe179, while UTX recognizes H3P30 by Pro1144 at different positions. **e** A superposition of JMJ13-NOG-H3K27me3 complex (in green) and JMJ14-NOG-H3K4me3 complex (in magenta, PDB code: 5YKO) shows almost identical overall structures. **f** A detailed view of the comparison of the zinc finger domains of JMJ13 (in green) and JMJ14 (in magenta). The two zinc finger domains possess similar overall structures with the first zinc ions occupying the same position but the second zinc ions located at different positions. **g** The detailed structures of the zinc coordination of two zinc ions from JMJ13 (upper panel, in green) and the JMJ14 (lower panel, in magenta) show that the first zinc ions have the same coordination but the second ones have different coordination. **h** The schematic representation of the sequence motif and coordination of the zinc ions of the JMJ13 C4HCHC domain (upper panel) and JMJ14 C5HC2 domain (lower panel). The spacing residues are denoted as X

(Supplementary Fig. 5a), all of which increased in severity with increasing *JMJ13* expression levels (Supplementary Fig. 5b). The JMJ13-GFP seedlings showed phenotypes similar to H3K27me3 silencing-deficient mutants *lhp1*[34] or *REF6* over-expression plants, with reduced size of leaf surface cells (Supplementary Fig. 5c, d)[25].

Some H3K27me3 target genes, including *APETALA1* (*AP1*), *APETALA3* (*AP3*), *PISTILLATA* (*PI*), *AGAMOUS* (*AG*), and *SEPALLATA3* (*SEP3*), which are normally expressed in flowers, were ectopically activated in the *JMJ13ox* seedlings. (Supplementary Fig. 5e). Other genes, for example, *FT* and *SOC1*, which are expressed at low levels in 10-day-old seedlings, and the meristem function genes *KNOTTED-LIKE FROM ARABIDOPSIS THALIANA 1* (*KNAT1*) and *TOUCH 4* (*TCH4*), all of which are targets of REF6, were also upregulated in *JMJ13ox* seedlings (Supplementary Fig. 5e). Consistent with these changes in gene expression, H3K27me3 but not H3K4, H3K9, and H3K36 showed a strong global reduction in the two strong *JMJ13ox* transgenic lines (Supplementary Fig. 5f).

**JMJ13 represses flowering**. We identified *jmj13*, a T-DNA insertion in the second intron of *JMJ13* (GABI116B03, At5g46910) (Supplementary Fig. 6a)[35]. No *JMJ13* transcript was detected in the *jmj13* mutant (Supplementary Fig. 6b–c). The plant Polycomb-group (Pc-G) protein CURLY LEAF (CLF), which functions as an H3K27me3 methyltransferase, is required to repress targets such as *AG* and *SHOOTMERISTEMLESS* (*STM*). CLF, thereby controls flowering time, leaf morphology, and floral organogenesis[36,37]. The *jmj13 clf* double mutant phenotype was similar to but weaker than that of *clf* in terms of flowering time, curling of leaves (Supplementary Fig. 6d–g) and fertility, indicating that *jmj13* partially suppresses the *clf* phenotype. Consistently, the expression levels of some H3K27me3 target genes that were ectopically activated in *clf* were reduced in the *jmj13 clf* double mutant (Supplementary Fig. 6h). However, the expression of *SEP3* and *AG* is not reduced in *jmj13 clf* relative to *clf*, although the leaf curling and early flowering phenotypes are weakly suppressed in the *jmj13 clf* double mutant. This could be due to other H3K27me3 target genes, or due to functional redundancy with REF6, ELF6 or other possible H3K27me3 demethylases. Together, these genetic interactions demonstrate that JMJ13 functions as an H3K27me3 demethylase that partially antagonizes the H3K27 methyltransferase CLF in vivo, although further investigation would be required to reveal the details of crosstalk between CLF and JMJ13 and whether the phenotypic suppression is due to direct antagonistic function at particular loci.

We further measured flowering time and scored total leaf number at bolting, in different photoperiod conditions (LD and SD) and under different ambient temperature conditions (low, 16 °C; and high, 28 °C). The *jmj13* mutants displayed early flowering in LD conditions[38], regardless of low or high

temperatures. However, *jmj13* plants flowered early at 28 °C, but not at 16 or 22 °C, when grown in SD conditions (Supplementary Fig. 7).

To verify that loss of JMJ13 function is responsible for the early flowering phenotype of *jmj13*, we transformed *jmj13* with a 7.0 kb genomic construct (gJMJ13) including 2.6 kb upstream and 0.5 kb downstream of the coding region. All the transgenic plants exhibited comparable flowering time to Col plants at 22 °C in LD conditions (Fig. 5a). Compared with the fully rescued *pJMJ13: JMJ13-HA jmj13* lines, lines containing a construct with point mutations in the two conserved iron-binding amino acids, H293A and E295A (*pJMJ13:JMJ13mu-HAjmj13*), showed similar flowering time to *jmj13* mutants. This suggests that the enzyme activity is necessary for JMJ13 function, together with the jumonji and zinc finger domains (Fig. 5a, b).

In Col plants, LD conditions and elevated ambient temperatures promote flowering. However, when grown in SD conditions, increased temperature could, to some extent, overcome the unfavorable day-length condition to promote flowering. Consistent with these phenotypic observations, the reproductive development of *jmj13* mutants was hypersensitive to, and promoted by, both LD conditions and elevated ambient temperatures (Fig. 5b). Collectively, these results suggest that JMJ13 negatively modulates flowering time under LD and in SD under higher temperature.

**Crosstalk between JMJ13 and other flowering time regulators**. To investigate the mechanism underlying JMJ13′s regulation of flowering in response to temperature, we crossed *jmj13* with *flm* and *svp*. The *flm jmj13* and *svp jmj13* double mutants flowered earlier in all conditions, including 22 °C and SD, whereas *jmj13* showed a wild-type flowering phenotype in 22 °C and SD (Supplementary Fig. 8a). Over-expression of FLM-β but not FLM-δ repressed early flowering in *jmj13* (Supplementary Fig. 8b–d). These genetic data are consistent with the possibility that FLM and SVP may act downstream or in parallel with JMJ13 in the regulation of flowering.

In the photoperiodic pathway, the circadian clock-regulated GIGANTEA (GI) protein positively regulates the oscillating expression of CO[2,5]. The transcript level and protein activity of CO are coordinately controlled by the light signaling pathway and the circadian clock. *FT* expression is activated by CO and repressed by FLM and SVP. Thus, transcriptional regulation of *FT* is a key output resulting from integration of photoperiodic cues with temperature signals. The flowering time of double mutants of *ft jmj13*, *co jmj13*, and *gi jmj13* were assessed by counting rosette and cauline leaf numbers in bolting seedlings (Supplementary Fig. 9a±d).

We found that all three double mutant lines displayed similar rosette leaf numbers to the *ft*, *co*, and *gi* single mutants (Supplementary Fig. 9a–d), and *FT* expression was upregulated in

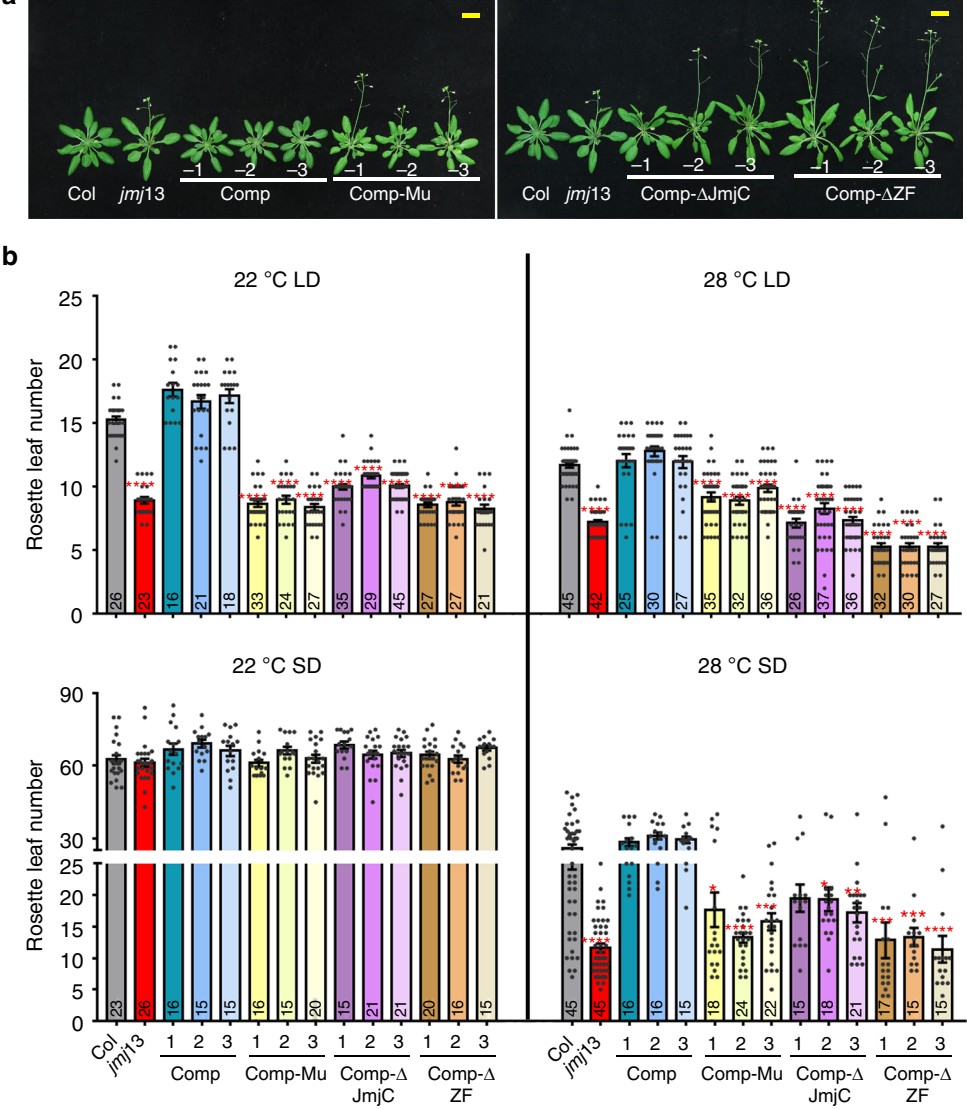

**Fig. 5** JMJ13 negatively modulates flowering time in a temperature- and photoperiod-dependent manner. **a** Three complementary lines Comp −1, −2 and −3 (p*JMJ13:JMJ13*-HA*jmj13*) showed similar flowering time with wild type in 22 °C LD conditions, but the complementary lines Comp-Mu −1, −2 and −3 with site mutation in the two conserved iron-binding amino acids (p*JMJ13:JMJ13mu*-HA*jmj13*), the complementary lines with truncation in JmjC Comp-ΔJmjc −1,−2, −3 (p*JMJ13:JMJ13*-Δ*Jmjc*-HA*jmj13*) and zinc finger domain Comp-ΔZF −1, −2, −3 (p*JMJ13:JMJ13*-Δ*ZF*-HA*jmj13*) remain early flowering. The plant images were created by the authors in this research. **b** The flowering time of complementary lines (Comp −1, −2, −3); (Comp-Mu −1, −2, −3); (Comp-ΔJmjc −1, −2, −3); (Comp-ΔZF 1, −2, −3) were assessed by counting rosette leaf numbers in bolting seedlings grown under 22 °C, 28 °C LD, and 22 °C, 28 °C SD conditions. Values are means ± SE of indicated number of plants for three independent biological repeats. *N* number was marked in the bottom of the column. Student's *t* tests was used to calculate the *P* value between Col and other lines. ****$P$ value < 0.0001; ***$P$ < 0.001; **$P$ < 0.01; *$P$ value < 0.05; Bar = 1 cm. The individual data points are shown as dots. Source data are provided as a Source Data file

*jmj13* in all early flowering conditions (Supplementary Fig. 9e). The mutations of *CO*, *GI*, and *FT* repressed early flowering in *jmj13*. These genetic interactions indicate that JMJ13 may crosstalk with GI/CO in flowering time regulation.

Furthermore, we performed RNA-seq for Col and *jmj13* grown under 22 °C LD conditions. The expression of *SVP* decreased in *jmj13* compared with wild type (Supplementary Fig. 10a, b), which is consistent with the genetic data. By contrast, the expression of *CO*, *Late Elongated Hypocotyl* (*LHY*) and *Circadian Clock Associated 1* (*CCA1*) was upregulated in *jmj13* (Supplementary Fig. 10a, b and Supplementary data 1), indicating that many genes are regulated directly or indirectly by additional mechanisms, such as circadian rhythm. Interestingly, we observed that high temperature and long-day photoperiod induced *JMJ13* expression and increase JMJ13

protein accumulation in 7-day-old seedlings (Supplementary Figs. 10c, d and 11). These results suggested that JMJ13 may affect flowering time through temperature- and photoperiod-mediated pathways.

We further analyzed the H3K27me3 levels in Col and *jmj13* by chromatin immunoprecipitation followed by sequencing (ChIP-Seq) in different day-length and temperature conditions. However, we did not observe the over-accumulation of H3K27me3 in *jmj13* at *FLM/SVP* and *CO/GI* and no significant differences between Col and *jmj13* were observed at these loci (Supplementary Fig. 12). This may due to the functional redundancy of ELF6, REF6 or other possible H3K27me3 demethylases. These observations are in consistent with recent report that global H3K27me3 levels were strongly elevated only in the triple *elf6 ref6 jmj13* mutant in Arabidopsis[38].

## Discussion

In most eukaryotic genomes, a large proportion of chromatin is enriched with H3K27me3. Erasure of these methyl groups is tightly controlled during development, and as part of acclimation to environmental conditions. Flowering is strictly regulated by various epigenetic factors. For example, we previously showed that the H3K4me3 demethylase JMJ14, and H3K27me3 demethylases REF6 and ELF6 directly regulate flowering integrators *FT*, *FLC*, and *SOC1*[22,25–27]. Recently, it was reported that JMJ13, REF6, and ELF6, shape the genome-wide distribution of H3K27me3 and control tissue-specific gene activation[38]. Global H3K27me3 levels were only increased in the triple *elf6 ref6 jmj13* mutant, conforming that JMJ13, REF6, and ELF6 are functional redundant. REF6 plays a major role in determining the distribution of H3K27me3, with ELF6 and JMJ13 apparently supplementing REF6 function[38], which partial explained that there is no over-accumulation of H3K27me3 in *jmj13*. During flowering, REF6 and ELF6 have distinct functions, as *ref6* and *elf6* mutants have late and early flowering phenotypes, respectively[21]. Here, we show that JMJ13 represses flowering in a temperature- and day-length-dependent manner in Arabidopsis. JMJ13's repressive function prevents precocious flowering under unfavorable environmental conditions. Although JMJ13, ELF6, and REF6 are all H3K27me3 demethylases, they each have different functions in Arabidopsis flowering control, indicating that their differential targeting lead to diverse pathways of flowering regulation. Further work would be needed to determine how JMJ13 affects flowering time, which JMJ13 targets that might be phenotypically relevant, and what are the differences and similarities among ELF6, REF6, and JMJ13.

Ambient temperature-dependent, flowering-time control is fine-tuned by multiple pathways in Arabidopsis. A distinct chromatin mark, H3K36me3 has been shown to affect temperature-induced alternative splicing, a major co-transcriptional/post-transcriptional regulatory mechanism to fine-tune ambient temperature-dependent flowering-time in Arabidopsis[39,40]. Promotion of flowering at higher temperatures has been proposed to counter the unfavorable SD photoperiod[41]. Here we show that another chromatin mark H3K27me3 can also regulate temperature- and photoperiod-dependent flowering regulation. The H3K27me3 demethylase JMJ13 may negatively modulate temperature-photoperiod compensation by dampening temperature-driven flowering-promotion in the absence of LD induction. LD photoperiod, and especially high temperature, induces *JMJ13* mRNA expression and JMJ13 protein accumulation. These results suggest that JMJ13 plays an important role in temperature- and photoperiod-dependent flowering time regulation.

## Methods

**Plant materials and growth conditions**. Arabidopsis plants used in the study were in the Columbia background. Primers for genotyping *jmj13* are listed in Supplementary Table 3. Plants were grown in growth chambers under LD conditions (16 h light/8 h dark) or SD conditions (8 h light/16 h dark) at 16 °C, 22 °C and 28 °C, respectively. The rosette leaves number were counted when the seedlings bolting. The mutants used in this study are *flm-3*[7], *svp-32* (SALK_072930), *ft-1* (CS56), *co-2* (CS55), *gi-3* (CS51), *elf6* (SALK_074694), *ref6* (SALK_001018), and *clf* (SALK_139371). The *lhp1* mutant used in this study was *tfl2-1*[42].

**Molecular cloning of JMJ13**. To clone the full-length *JMJ13* cDNA, we amplified a 2364-bp cDNA fragment from a reverse transcribed cDNA pool derived from Col-0 seedlings. The PCR product was cloned into pENTR/D-TOPO (Invitrogen). A Quik Change Site-Directed Mutagenesis Kit (Stratagene) was used to make the point mutations in JMJ13. The wild-type and mutant *JMJ13* coding sequences were introduced into the pEarley-Gate104 vector. Primers used for cloning are listed in Supplementary Table 3.

**Transient expression in Nicotiana benthamiana leaves**. The constructs for JMJ13, or JMJ13$^{H293A/E295A}$ fused with GFP were transformed into *Agrobacterium tumefaciens* cells (strain EHA105). These cells were then injected into *Nicotiana*

*benthamiana* leaves, which were harvested for nuclear isolation and immunostaining or immunoprecipitation after 48 h[27].

**In vivo histone demethylation assays**. Nuclei transfected with JMJ13-GFP or JMJ13$^{H293A/E295A}$-GFP mutants were visualized by observing the GFP signal under a fluorescence microscope. Immunolabeling was performed by using histone methylation-specific antibodies (H3K27me3, Millipore 07-449; H3K27me2, Millipore 07-452; H3K27me1, Millipore 07-448; H3K4me3, Millipore 07-473; H3K4me2, Millipore 07-030; H3K4me1, Millipore 07-436; H3K9me3, Millipore 07-442; H3K9me2, Millipore 07-441; H3K9me1, Millipore 07-450; H3K36me3, Abcam ab9050; H3K36me2, Millipore 07-274; H3K36me1, Millipore 07-548). All these antibodies were diluted by 1:100. Alexa Fluor 555 or 488-conjugated goat anti-rabbit (1:500, Invitrogen) were used as secondary antibodies to determine the specific lysine modification site in vivo. After incubation with secondary antibody, nuclei on the slide were mounted by one drop of VECTASHIELD Mounting Medium with 4′,6-diamidino-2-phenylindole (DAPI) (Vector Laboratories), then photographed under a fluorescent microscope (Olympus BX51). ImageJ (National Institutes of Health) was used for quantification of the immunolabeled nuclei.

**Transcript level analysis**. Total RNA was extracted using TRIzol Reagent (Invitrogen) for reverse transcription-PCR (RT-PCR) and real-time quantitative PCR (qPCR) analysis. Real-time qPCR analysis was performed using SYBR Green (CWBIO, CW0760A). Primers used for transcriptional analysis are listed in Supplementary Table 3.

**RNA sequencing analysis**. Total RNA was isolated from newly fertilized siliques of six weeks seedlings using a TRIzol kit (Invitrogen). Paired-end sequencing libraries with an average insert size of 400 bp were prepared with a TruSeq RNA Sample Preparation Kit v2 (Illumina) and sequenced on the HiSeq2500 (Illumina). Raw data obtained from Illumina sequencing were processed and filtered using the Illumina pipeline (http://www.Illumina.com) to generate FastQ files. Finally, about 8 Gb high-quality 150-bp paired-end reads were generated from each library. FastQC (http://www.bioinformatics.babraham.ac.uk/projects/fastqc/) was initially run to assess the overall quality of all sample reads. Poor quality bases were filtered out using Sickle with parameters "-mode pe; -t sanger–q 20 –l 50" (https://github.com/najoshi/sickle). The quality filtered reads were aligned to the Arabidopsis reference genome using TopHat2 version 2.0.9[43] with the parameters "-N 3–read-edit-dist 3–segment-mismatches 1 -p 20 -r 0 -g 20–microexon-search–b2-D 20–b2-R 3–no-coverage-search". HTseq software (http://www-huber.embl.de/users/anders/HTSeq/doc/overview.html) was used to count the number of reads mapped to each of the genes. DEGseq was used for differential expression analysis with the "Fisher's Exact Test" method[44]. The genes showing an absolute value of log2 (fold change; *jmj13* mutant/WT) ≥ 0.6 and adjusted *P* value (false discovery rate; FDR) <0.05 were considered as differentially expressed genes.

**ChIP sequencing**. A total of 2 g of 7-day-old seedlings sample was powdered in liquid nitrogen using a pestle and mortar and fixed with formaldehyde then immunoprecipitated with anti-H3K27me3 (Millipore 07-449), anti-H3 (Abcam; ab1791) antibodies. After decross-linking, proteinase K, and RNase treatment, the immunoprecipitated DNA was purified by phenol/chloroform extraction for the additional experiments including Illumina single-end sequencing[22,45]. ChIPed DNA was ligated with Illumina single-end genome sequencing adapters and then fragments fractionated, PCR amplified and sequenced according to standard protocols (single-end 36 cycles). ChIP-seq reads were aligned to Arabidopsis genome build TAIR10 by Bowtie 2[46] using default parameters with a local alignment model. Duplicated reads and low-mapping quality reads were identified and removed with SAMtools[47]. Enriched intervals were identified by MACS version 2.1.0[48] with default parameters. Density maps of reads for visualization were based on reads of the 200-bp extension of sequencing reads in the 3′-direction after total reads normalization[49].

**Western blot assay**. Ten-day-old seedlings were ground in liquid nitrogen and the powder was boiled for 5 min in protein sample buffer. The proteins were resolved on a 15% SDS/PAGE gel and transferred onto nitrocellulose membranes (Bio-Rad). Then the membranes probed with anti-H3K27me3 (Millipore 07-449); anti-H3K27me2 (Millipore 07-452); anti-H3K27me1 (Millipore 07-448); anti-H3K4me3 (Millipore 07-473); anti-H3K4me2 (Millipore 07-030); anti-H3K4me1 (Millipore 07-436); anti-H3K9me2 (Millipore 07-441); anti-H3K9me1 (Millipore 07-450); anti-H3K36me3 (Abcam ab9050); anti-H3K36me2 (Millipore 07-274); anti-H3K36me1 (Millipore 07-548); anti-H3 (Abcam ab1791) or anti-GFP (Roche 11814460001) in TBST (137 mM NaCl, 20 mM Tris·HCl pH 7.6, 0.1% Tween-20). After three washes with TBST, the signals were detected with Immobilon Western Chemiluminescent HRP Substrate (Millipore) for histone antibodies or Super Signal West Dura Extended Duration Substrate (Thermo Fisher Scientific) for GFP antibody.

**Protein expression and purification**. The catalytic fragment of Arabidopsis JMJ13 (JMJ13CD, residues 90–578) was cloned into a pET-Sumo vector to fuse an N-terminal hexahistidine plus yeast sumo tag. The plasmid was transformed into *E. coli* strain BL21(DE3) RIL and the transformants were cultured at 37 °C in LB

medium. When the $OD_{600}$ of cell culture reached 0.7, the protein expression was induced by adding IPTG to a final concentration of 0.2 mM and the cells were cooled to 20 °C. The recombinant expressed protein was purified using a HisTrap column (GE Healthcare). The hexahistidine plus yeast sumo tag was removed by ulp1 protease digestion followed by a second step HisTrap column (GE Healthcare). The target protein was further purified on Heparin and Superdex G200 columns (GE Healthcare). The untagged JMJ13CD easily precipitates in the in vitro activity assay. We further cloned JMJ13CD into a pMal vector (New England Biolabs) to fuse an MBP tag to the target protein. The MBP-tagged JMJ13CD was expressed in *E. coli* strain BL21(DE3) RIL with IPTG induction and purified using amylose resin (New England Biolabs), Heparin, and Superdex G200 columns (GE Healthcare). All the mutations were generated using a PCR based method and purified using the same protocol as wild-type protein. The chemicals and peptides were purchased from Sigma-Aldrich and GL Biochem Company, respectively.

**Crystallization and structure determination**. The purified JMJ13CD was concentrated to a final concentration of 10 mg ml$^{-1}$ and mixed with α-KG with a molar ratio of target at 1:4 at 4 °C for one hour. The crystal screening was carried out using sitting-drop vapor diffusion method at 4 °C. The JMJ13-α-KG complex was crystallized in 0.1M MES, pH 6.5, 8% dioxane, and 1.6 M ammonium sulfate. To get crystals of the JMJ13–NOG–H3K27me3 complex, we first grew crystals of JMJ13–NOG in the same conditions as JMJ13–α-KG, and then soaked the crystals with 10 mM H3(24–35)K27me3 peptide for 60 h. All the diffraction data were collected at beamline BL19U1 of the National Center for Protein Sciences Shanghai (NCPSS) at the Shanghai Synchrotron Radiation Facility (SSRF) and processed using the program HKL2000/3000 package[50]. A summary of the data collection statistics is listed in Supplementary Table 2.

The structure of the JMJ13-α-KG complex was determined using the SAD method with the anomalous signal of the zinc peak as implemented in the program Phenix[51]. The model building and structure refinement were conducted using the programs Coot and Phenix, respectively[51,52]. Throughout the refinement, the geometry of the structure was monitored using the program Molprobity[53]. The structure of the JMJ13-NOG-H3K27me3 complex was solved using the molecular replacement method with the structure of the JMJ13-α-KG complex as the search model and was refined using the same protocol as used for the JMJ13-α-KG complex. Density modification was applied with the program SOLOMON[54]. A summary of the structure refinement statistics is listed in Supplementary Table 2.

**In vitro histone demethylation assay**. Since the untagged JMJ13CD easily precipitates during the in vitro assay, we used MBP-tagged JMJ13CD in the assay to increase the stability of the protein. The purified MBP-tagged JMJ13CD or its mutants (10 μM) were incubated with the H3(20–34)K27me3 peptide (80 μM) in a reaction buffer with 80 μM Fe(NH₄)₂(SO₄)₂, 2 mM ascorbic acid, 1 mM α-KG, 50 mM Tris-HCl pH 7.3, and 150 mM NaCl at 25 °C. The reaction solution was incubated at 25 °C for 2 h and then was quenched by heating to 95 °C for 3 mins. The reaction mixture was further desalted by ZipTip (Millipore). The eluted peptides were spotted on the MALDI plate and cocrystallized with 10 mg ml$^{-1}$ α-cyano-4-hydroxycinnamic acid in 60% acetonitrile with 0.1% trifluoroacetic acid (TFA). Then, a 5800 MALDI-TOF/TOF mass spectrometer (ABsciex, Foster City, CA) was used for analyzing the samples. For quantification, peptide mass spectra were recorded in reflector mode from triplicate reactions with each spectrum acquired from 1250 laser shots. The resulting mass spectra were processed by the Data Explorer software.

**Reporting Summary**. Further information on experimental design is available in the Nature Research Reporting Summary linked to this article.

## Data availability

X-ray structures have been deposited in the RCSB Protein Data Bank with the accession codes: 6IP0 for the JMJ13–α-KG complex and 6IP4 for the JMJ13–NOG–H3K27me3 complex. RNA-Seq and ChIP-Seq data have been uploaded to NCBI SRA with accession number SRP168443 and SRP174856, respectively. The Source Data underlying Figs. 1, 3, 5, Supplementary Figs. 1, and 5–10 are provided as a Source Data file.

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

## Acknowledgements

We thank staff at beamline BL19U1 of the National Center for Protein Sciences Shanghai (NCPSS) at the Shanghai Synchrotron Radiation Facility (SSRF) for data collection, the Proteomic Core Facility of Shanghai Center for Plant Stress Biology for assistance with the mass spectrometry experiment, and Markus Schmid for providing the seeds of *flm-3* and the FLM-β and FLM-δ transgenic seeds. This work was supported by National Natural Science Foundation of China (31788103 to X. Cao, 31501029 and 31870243 to S.Z., and 31622032 to J.D.), the Chinese Academy of Sciences (Strategic Priority Research Program XDB27030201 and QYZDY-SSW-SMC022 to X. Cao and J.D.), the Guangdong Innovation Research Team Fund (2016ZT06S172) to S.L., the Advanced Talents Foundation of Hebei Education Department (GCC2014002) to D.S., and the National Key Laboratory of Plant Molecular Genetics.

## Author Contributions

S.Z., H.H., H.R., Z.L., Q.Q., W.Q., X.L., X.Chen, X.Cui, S.L., and B.Z. performed the experiments and analyzed the data. S.Z., D.S., X.Cao, and J.D. conceived the study, designed the experiments, and wrote the article.

## Additional information

**Competing interests:** The authors declare no competing interests.

