## [Peer Review File · Nature Communications]

Reviewers' comments:

Reviewer #1 (Remarks to the Author):

The manuscript by Zheng et al. describes the structure, activity, specificity of JMJ13 as well as its biological function in flowering time regulation. In vitro and in vivo experiments showed that JMJ13 specifically demethylated H3K27me3 and not other methylated histones tested here. The substrate recognition mechanism was revealed by structural analysis of JMJ13CD and the H3K27me3 peptide, which showed that Phe179 interacted with H3P30, thus distinguishing H3K27me3 from H3K9me3. In addition, overexpression of JMJ13 led to phenotypes resembling those seen in polycomb mutants, and *jmj13* partially suppressed the *clf* phenotype. These biochemical, structural and genetic analyses are seen as a particular strength as they clearly demonstrated that JMJ13 is a H3K27me3 demethylase.

With regard to the biological functions of JMJ13, the authors found that *jmj13* displayed an interesting defect in flowering time regulation: while *jmj13* was early flowering at any temperature under long day conditions, it was only early flowering at high temperature under short day conditions. Results from genetic analyses suggested that JMJ13 may act upstream of both the SVP-FLM and the GI-CO pathways. This phenotype is highly novel and potentially very important for many crop species. Hopefully the direct target gene(s) of JMJ13 can be identified in future studies.

My main concern is that the defects in H3K27me3 in neither JMJ13ox or *jmj13* were not described. *jmj13* was early flowering under some conditions but fairly normal otherwise, indicating that H3K27me3 changes in *jmj13* may be mild, specific to a few genes, or occur predominantly in certain cell types, making it difficult to detect H3K27me3 changes. However, the authors are encouraged to discuss this directly, instead of avoiding it in the manuscript, as readers will undoubtedly wonder.

Reviewer #2 (Remarks to the Author):

In this manuscript Zheng et al. report studies on the Arabidopsis JMJ13 putative histone demethylase. They demonstrate its demethylase activity both in vitro and in vivo and solve the crystal structure revealing the molecular basis of its specificity. Finally, they demonstrate its role as a flowering repressor. Overall this is a thorough study that lends both mechanistic and functional insight into this plant demethylase, and should be of general interest. Some points to be addressed are noted below:

- 1) General English editing is needed throughout the text.
- 2) It is overall very hard to keep track of all of the plant and mammalian demethylase families and specific demethylases referenced throughout the text. A table summarizing all these would be nice.
- 3) In addition, a domain architecture comparison and sequence alignment would go a long way to help the reader with all of the comparisons between families. As a main point of the paper is that JMJ13 is unique in that it belongs to the KDM4 family but resembles the KDM5 family, yet has activity like the KDM6 family, the differences and similarities between families need to be very clear. Currently it is somewhat confusing in how it is presented and should be substantially clarified. (Though I note that figure 7 is very nice.)
- 4) A couple of minor points....Stating that methylation is one of the "most important" post-translation modifications is a vague statement at best, please re-word. Please remove references to "as previously described" and add experimental details.

Reviewer #3 (Remarks to the Author):

The paper describes the characterization of the Arabidopsis histone demethylase JMJ13 and has three main aspects: first, the in vivo and in vitro assays to show that JMJ13 acts as an H3K27me3 histone demethylase (HDM); second, structure of the JMJ13 catalytic domain and some structure/function assay involving mutation of key residues and assay in vivo or in some cases in planta; thirdly, phenotypic characterization of JMJ13 mutants and their role in flowering time regulation. Overall I found it an interesting paper that was well written and of broad interest for the epigenetics community. Part one has been covered elsewhere, as the Kaufman group recently published a Nature Plants paper with detailed genomic analysis of various H3K27me3 HDMs (REF6, ELF6, JMJ13) and show that JMJ13 acts as a H3K27me3 HDM, as the authors correctly acknowledge/cite. The current study adds in that it presents in vitro assays with purified JMJ13 catalytic domain protein based on mass spectrometry and this is important I think as the in vivo assays (based on transgene expression in tobacco) don't entirely rule out that some other (tobacco) proteins are involved, and also assay of total histone extracts from plant over-expressing JMJ13. Part two, the structural part I lack expertise to comment on in detail but to me seemed the most interesting part of the paper, and adds in that the structure suggests how phylogenetically distinct plant and animal H3K27me3 HDM may have converged upon similar specificities. Part three seems incomplete and the weakest part of the paper. That jmj13 mutants affect flowering time and have different effects at different temperatures in short days was adequately shown, but the claims that this is due to jmj13 acting via CO/GI and SVM/FLM seemed incomplete and less convincing, the interpretations are based on genetics but with no supporting data for example looking at expression of these genes in mutant vs wild type or at chromatin status (H3K27me3 methylation etc). One option might be to omit much of part three (and publish later when more complete) as parts one and two to my view make an interesting story, perhaps supplemented with the JMJ13 over expression data, alternatively to revise to improve/strengthen part three. More specific comments as follows:

Line 58 – this should cite some of the primary research papers from the Coupland group.

Line 98 – 99 and elsewhere – probably my ignorance but I found the zinc finger nomenclature hard to follow and some further explanation of what the various C4HCHC etc refer to could be helpful for a general audience given that the structural part of the results section has a lot about the zinc fingers.

Line 219 – Text goes from Supp Fig 4 to Supp Fig 6 with no mention of S5, i.e. figure numbering and order needs attention.

Line 227 – analysis of jmj13 clf double mutants. The phenotypic data suggest that the jmj13 mutant gives a weak suppression of the clf mutant phenotype, as has been shown for eg ref6 mutants. However, the Q RT PCR data of target gene expression does not really support this. Three genes have been shown to be causal for the leaf curling and early flowering of clf mutants (SEP3, AG, FT). Expression of SEP3 and AG is not reduced in jmj13 clf relative to clf, and FT expression is not analysed. It is not clear what tissue is being sampled i.e. leaves or whole seedlings, what age, how grown etc. Panel A also includes jmj13 clf swn/+ mutants, where suppression does not occur, but this is not referred to or explained in the text. Some analysis of H3K27me3 at targets could also be helpful.

Line 238 – 7.0 kb rather than 7.0 Mb intended here?

Line 242 refers to H293 and E295 as iron binding, yet the structural figures (Fig 3 C) shows these

residues binding Nickel nor Iron ions which is slightly confusing. Perhaps either Ni²⁺ or Fe²⁺ will do but clarification is needed.

Line 260 – Authors suggest that FLM and SVP act downstream of JMJ13 based on experiments over expressing e.g. SVP transgene in *jmj13* mutant background and overcoming the early flowering. This implies they act in the same pathway, but I think other explanations are possible, e.g. SVP acts in parallel on a common target. No analysis of SVP or FLM expression in *jmj13* mutants is provided, no ChIP either of H3K27me₃ levels or JMJ13 binding so it seems incomplete.

Line 275 – again, more is needed to support that *jmj13* acts via CO and GI, the genetics is consistent with this but on its own not compelling, expression analysis etc would help.

Line 280 – I think this should be KDM6 not KDM4?

Line 288 – This part and Figure 7 should be presented in results rather than first mentioned in discussion.

Reviewer #4 (Remarks to the Author):

In this work, Zheng et al. determined in the one hand the structure of the JMJ13 H3K27me₃ demethylase catalytic domain-peptide complex, and in parallel, they also investigated the role of JMJ13 in flowering time control.

They demonstrated that JMJ13 possesses H3K27me₃ site-specific demethylase activity both in vitro and in vivo, as it was reported previously for this protein (Yan et al., 2018) and for two other H3K27me₃ demethylases belonging to the same KDM4 subfamily, ELF6 and REF6, what diminish the novelty of these results.

They further determined the crystal structures of JMJ13 in both H3K27me₃ peptide-free and bound forms. They concluded that JMJ13 possesses a unique C4HCHC-type zinc finger, and not the previously predicted C5HC2-type zinc finger domain, despite the two zinc finger types share a similar folding topology. Interestingly, they found that the overall structure of JMJ13 catalytic domain resembles the previously reported structures of human KDM5A/B/C and Arabidopsis JMJ14. They also provided detailed structural insight into the substrate specificity of JMJ13 showing that the substrate H3K27me₃ peptide is specifically recognized by hydrogen bonding and hydrophobic stacking interactions. Besides, and to dissect the catalytic mechanism, they performed a thorough structure-based mutagenesis study, determining key residues for the activity of JMJ13. Nevertheless, further functional analyses in planta will be required to unravel how this putative plant KDM4 demethylase may use the KDM5 subfamily like-domain architecture to carry out the molecular functions of animal KDM6 demethylases.

To investigate the biological function of JMJ13, they produced several Arabidopsis transgenic lines overexpressing JMJ13, which show a strong global reduction in H3K27me₃ levels and display pleiotropic defects, including acceleration of flowering time; in parallel, they analyzed Arabidopsis loss of function *jmj13* mutants, which also flower earlier, but do not show clear changes in the global H3K27me₃ levels, possibly due to the redundancy with other H3K27me₃ demethylases. That JMJ13 plays a role in flowering time control was already reported in an independent study by Yan et al., 2018, what again compromise at certain degree the novelty of some of the data presented in this manuscript.

Interestingly, Zheng et al. reveal that JMJ13 modulates flowering time in a temperature and day-length-dependent manner. They showed that impaired JMJ13 function leads to early flowering in both LD and SD conditions under warm temperature, but not in SD conditions with low temperature. They also provided genetic evidences suggesting that JMJ13 is a flowering repressor, acting upstream of both GI-CO and SVP-FLM pathways. Besides, the genetic interactions observed

between the H3K27me3 methyltransferase CLF and JMJ13 demonstrate that the latter functions as an H3K27me3 demethylase that antagonizes CLF *in vivo*.

However, the main drawback to this part of the paper is that there is not very much mechanistic insight for the putative role of JMJ13 in the temperature- and photoperiod dependent flowering regulation. I consider that these observations present an exciting start to answering the role of JMJ13 as an epigenetic repressor of flowering time, but clearly more work is needed to unveil the molecular mechanisms mediated by JMJ13 in the ambient temperature and photoperiod-dependent control of this developmental transition.

Major points

1. What is the pattern of expression for JMJ13? (at tissue and temporal levels or under different environmental conditions of temperature and photoperiod?). There is already some information published in a previous report (Yan et al., 2018). Once established these patterns, the profile of specific gene regulation by JMJ13 under particular growing conditions of temperature and photoperiod should be explored by following RNA-seq approaches with the *jmj13* mutant.
2. Which are the direct targets of JMJ13 in flowering time control? The identification of genome wide binding sites of JMJ13 by ChIP-seq approaches involving the HA-tagged JMJ13 complemented plants shown in Fig.4, will be highly recommended to unveil putative target genes of this protein (or at least they should demonstrate by ChIP-PCR the direct binding of JMJ13 to a number of putative targets related to flowering time control that are misregulated in the *jmj13* mutant).
3. It would be also highly informative to provide insight on the mechanisms of JMJ13 target site recruitment given that apparently JMJ13 has no DNA-binding domain.
4. Temporal dynamics of JMJ13 –mediated H3K27 demethylation at genome wide level (or at least in a number of putative targets related to flowering time control) should be analyzed under different conditions of temperature and photoperiod in the *jmj13* mutant.
5. The *elf6* and the *jmj13* loss-of-function mutants displayed early flowering phenotypes while *ref6* mutants flower later than WT, suggesting that these H3K27me3 demethylases can influence flowering via different pathways. However, it would be interesting to check the possible contribution of ELF6 and REF6 in the temperature- and photoperiod dependent flowering control by analyzing the flowering behaviour of *elf6*, *ref6* or double and triple mutant combinations. Also it may be worth to assess the flowering behaviour of JMJ13ox and *clf* mutants under the same experimental conditions.
6. JMJ13ox plants displayed dwarf seedlings, early flowering and upward curling leaves pleiotropic phenotypes. Loss of function *jmj13* mutants are also early flowering. The authors should discuss this observation, where the loss- and the gain-of-function of JMJ13 cause the same flowering phenotype. Besides, the genetic interaction observed for JMJ13 and CLF in the control of flowering time, where the double mutant *jmj13clf* flowers later than any of the *jmj13* or *clf* single parentals, that are both early flowering mutants, and similarly to WT plants (Suppl. Figure 5d), should be explained in more detail. Same for the genetic interaction revealed for *jmj13clf* *swn* +/- plants regarding flowering time that it is shown in Suppl. Figure 5d and that it is not even described in the manuscript.

Minor points:

1. Indicate the *svp* allele used in the studies performed in Fig.5.
2. It would be highly informative to include FT and FLC expression data in the analysis performed in Suppl. Fig. 5 e, involving *jmj13* and combinations with *clf* mutation.
3. Writing throughout the manuscript should be revised. A number of suggestions are provided below, but additional editing is strongly advised.

Lane 51 long-day should be "LD"

Lane 68-69. Please rephrase to: ... in a temperature –dependent manner, and a strong binding of PIF4 to FT depends on the eviction of H2A.Z nucleosomes

Lane 123 MOLDI should be changed by "MALDI"

Lane 141 Add "was determined" after resolution.

Lane 161 Change "locating" by "located".

Lane 230 Change "antagonist" by "antagonize".

Lane 230 Change "methyltransferases" by "methyltransferase".

Lane 252. Please rephrase to: JMJ13 negatively modulates flowering time under LD and in SD

under high temperatures.

Lane 360 Change "Colombia " by "Columbia "

Lane 634-635: Rephrased to: "key residues involved in peptide binding and catalysis is decreasing the activity...."

Lane 680: Rephrased to: Fig 6. The role of JM13 in photoperiod –regulated flowering...

Several Col all over the text are in italics and should not be.

Suppl Fig 4 panel f: *jmj* mutation should be written in lower case and italics

Suppl Fig 5 panel d: include *jmj13clfswn*+/- in the figure legend and describe the result observed for the flowering time of this plant in the manuscript text.

We thank all the reviewers for their positive and constructive comments. We have improved our manuscript according to the reviewers' suggestion and made a point-to-point response as below with the reviewers' comments and our response in regular and red, respectively. In the main text, all the changes are also marked by red.

Reviewers' comments:

Reviewer #1 (Remarks to the Author):

The manuscript by Zheng et al. describes the structure, activity, specificity of JMJ13 as well as its biological function in flowering time regulation. In vitro and in vivo experiments showed that JMJ13 specifically demethylated H3K27me3 and not other methylated histones tested here. The substrate recognition mechanism was revealed by structural analysis of JMJ13CD and the H3K27me3 peptide, which showed that Phe179 interacted with H3P30, thus distinguishing H3K27me3 from H3K9me3. In addition, overexpression of JMJ13 led to phenotypes resembling those seen in polycomb mutants, and *jmj13* partially suppressed the *clf* phenotype. These biochemical, structural and genetic analyses are seen as a particular strength as they clearly demonstrated that JMJ13 is a H3K27me3 demethylase.

With regard to the biological functions of JMJ13, the authors found that *jmj13* displayed an interesting defect in flowering time regulation: while *jmj13* was early flowering at any temperature under long day conditions, it was only early flowering at high temperature under short day conditions. Results from genetic analyses suggested that JMJ13 may act upstream of both the SVP-FLM and the GI-CO pathways. This phenotype is highly novel and potentially very important for many crop species. Hopefully the direct target gene(s) of JMJ13 can be identified in future studies.

We thank the reviewer for the positive comments.

My main concern is that the defects in H3K27me3 in neither JMJ13ox or *jmj13* were not described. *jmj13* was early flowering under some conditions but fairly normal otherwise, indicating that H3K27me3 changes in *jmj13* may be mild, specific to a few genes, or occur predominantly in certain cell types, making it difficult to detect H3K27me3 changes. However, the authors are encouraged to discuss this directly, instead of avoiding it in the manuscript, as readers will undoubtedly wonder.

We thank the reviewer for the constructive comments. Indeed, we only described the global H3K27me3 changing in JMJ13ox lines but not in *jmj13* mutant. In this version, we have added our detection of H3K27me3 changes in *jmj13* in the text.

We have also tried to compare H3K27me3 levels in Col and *jmj13* by chromatin immunoprecipitation followed by sequencing (ChIP-Seq). However, the H3K27me3 changes in *jmj13* are mild and no significant changes were detected in *FLM* and *SVP* as well as in *CO* and *GI*. We think this is likely due to the functional redundancy of ELF6, REF6 and other H3K27me3 demethylases. Alternatively, the changes may occur specifically to a few genes, or occur predominantly in certain cell types, making it difficult to detect H3K27me3 changes in *jmj13* mutants. We added this discussion in the manuscript.

Reviewer #2 (Remarks to the Author):

In this manuscript Zheng et al. report studies on the Arabidopsis JMJ13 putative histone demethylase. They demonstrate its demethylase activity both in vitro and in vivo and solve the crystal structure revealing the molecular basis of its specificity. Finally, they demonstrate its role as a flowering repressor. Overall this is a thorough study that lends both mechanistic and functional insight into this plant demethylase, and should be of general interest. Some points to be addressed are noted below:

We thank the reviewer for the positive comments.

1) General English editing is needed throughout the text.

We have improved the English with the help of a native speaker and professional editor.

2) It is overall very hard to keep track of all of the plant and mammalian demethylase families and specific demethylases referenced throughout the text. A table summarizing all these would be nice.

As suggested by the reviewer, we have added Supplementary Table 1, listing the representative enzymes of human and Arabidopsis KDM4/KDM5/KDM6 family histone demethylases, along with their substrates and their domain architectures.

3) In addition, a domain architecture comparison and sequence alignment would go a long way to help the reader with all of the comparisons between families. As a main point of the paper is that JMJ13 is unique in that it belongs to the KDM4 family but resembles the KDM5 family, yet has activity like the KDM6 family, the

differences and similarities between families need to be very clear. Currently it is somewhat confusing in how it is presented and should be substantially clarified. (Though I note that figure 7 is very nice.)

Plant jumonji histone demethylases are grouped by sequence alignment of their JmjC domains (Lu et al.,2008). Arabidopsis JMJ13, REF6 and ELF6 are grouped into the KDM4 subfamily due to their sequence similarity to the JmjC domains of human KDM4A/4B/4C. The catalytic fragments of JMJ13 as well as KDM5 family proteins possess a jumonji-helical-zinc finger triple linked domain architecture. In contrast, the other two members of Arabidopsis KDM4 family, ELF6 and REF6, each have four canonical DNA-binding C2H2 zinc finger domains at the end of C-terminus, which is quite different from the domain architecture of JMJ13. Therefore, we think that JMJ13 is an evolutionary side branch in KDM4 family, with similar domain architecture of KDM5 family proteins and performs the function of H3K27me3 demethylases similar to KDM6 family proteins. Because of the almost totally different domain architecture, we think it may be quite hard to get sufficient information from the sequence alignment. Instead, as suggested by the reviewer, we have listed and compared the domain architectures of KDM4/5/6 from human and Arabidopsis in the new Supplementary Table 1 and rewritten this part to clarify this point.

4) A couple of minor points....Stating that methylation is one of the “most important” post-translation modifications is a vague statement at best, please re-word.

We have rewritten this sentence as suggested.

Please remove references to “as previously described” and add experimental details.

We have removed the phrase “as previously described” and added the details.

Reviewer #3 (Remarks to the Author):

The paper describes the characterization of the Arabidopsis histone demethylase JMJ13 and has three main aspects: first, the in vivo and in vitro assays to show that JMJ13 acts as an H3K27me3 histone demethylase (HDM); second, structure of the JMJ13 catalytic domain and some structure/function assay involving mutation of key residues and assay in vivo or in some cases in planta; thirdly, phenotypic characterization of JMJ13 mutants and their role in flowering time regulation. Overall I found it an interesting paper that was well written and of

broad interest for the epigenetics community. Part one has been covered elsewhere, as the Kaufman group recently published a Nature Plants paper with detailed genomic analysis of various H3K27me3 HDMs (REF6, ELF6, JMJ13) and show that JMJ13 acts as a H3K27me3 HDM, as the authors correctly acknowledge/cite. The current study adds in that it presents in vitro assays with purified JMJ13 catalytic domain protein based on mass spectrometry and this is important I think as the in vivo assays (based on transgene expression in tobacco) don't entirely rule out that some other (tobacco) proteins are involved, and also assay of total histone extracts from plant over-expressing JMJ13. Part two, the structural part I lack expertise to comment on in detail but to me seemed the most interesting part of the paper, and adds in that the structures suggests how phylogenetically distinct plant and animal H3K27me3 HDM may have converged upon similar specificities. Part three seems incomplete and the weakest part of the paper. That jmj13 mutants affect flowering time and have different effects at different temperatures in short days was adequately shown, but the claims that this is due to jmj13 acting via CO/GI and SVM/FLM seemed incomplete and less convincing, the interpretations are based on genetics but with no supporting data for example looking at expression of these genes in mutant vs wild type or at chromatin status (H3K27me3 methylation etc). One option might be to omit much of part three (and publish later when more complete) as parts one and two to my view make an interesting story, perhaps supplemented with the JMJ13 over expression data, alternatively to revise to improve/strengthen part three. More specific comments as follows:

We thank the reviewer for the overall positive comments. We are glad that the reviewer is satisfied with the Part 1 and 2. In Part 3 of the manuscript, we characterize JMJ13 from biochemistry-structure to genetics to shape a more complete story to study JMJ13 more comprehensively. Therefore, we prefer to keep Part 3 in our manuscript. As suggested by the reviewer, we have strengthened Part 3 by adding RNA-seq analysis (Fig. 8a and Supplementary data 1), JMJ13 expression pattern in different growth conditions in Fig. 8c and d and Supplementary Figure 8, and ChIP-seq in Supplementary Figure 9. High temperature and long-day photoperiod, especially high temperature, could induce mRNA expression and accumulation of higher levels of JMJ13. These results suggest that JMJ13 plays an important role in temperature- and photoperiod-dependent flowering time regulation. We have added the RNA-seq, expression pattern and ChIP-seq related part in the results section of the main text and add new figures and data (Fig. 8; Supplementary 8 and 9; Supplementary data 1) correspondingly.

Line 58 – this should cite some of the primary research papers from the Coupland group.

As suggested by the reviewer, we have added the primary research papers from Coupland's group.

Line 98 – 99 and elsewhere – probably my ignorance but I found the zinc finger nomenclature hard to follow and some further explanation of what the various C4HCHC etc refer to could be helpful for a general audience given that the structural part of the results section has a lot about the zinc fingers.

The zinc ion (Zn^{2+}) of a single zinc finger motif is coordinated by four amino acids, which can either be a cysteine(C) or a histidine (H). Zinc fingers are often named following the occurrence of zinc-coordinating cysteines and histidines in the primary amino acid sequence. For example, a C2H2-type zinc finger has two coordinating cysteine residues and two coordinating histidine residues (from N to C terminus) in the amino acid sequence. JMJ13 as well as proteins from KDM5 family, are predicted from the primary sequence to have a C5HC2-type zinc finger motif (5 cysteine residues followed by a histidine and 2 additional cysteine residues). However, for JMJ13, the zinc coordination is slightly different from KDM5, in that a different histidine residue is actually coordinating and substitutes for the predicted cysteine in the zinc finger motif, resulting in a C4HCHC-type zinc finger. We have added some general explanations in our revised manuscript as suggested by the reviewer.

Line 219 – Text goes from Supp Fig 4 to Supp Fig 6 with no mention of S5, i.e. figure numbering and order needs attention.

We have rearranged the text and Supp Fig. orders.

Line 227 – analysis of *jmj13* *clf* double mutants. The phenotypic data suggest that the *jmj13* mutant gives a weak suppression of the *clf* mutant phenotype, as has been shown for eg *ref6* mutants. However, the Q RT PCR data of target gene expression does not really support this. Three genes have been shown to be causal for the leaf curling and early flowering of *clf* mutants (*SEP3*, *AG*, *FT*). Expression of *SEP3* and *AG* is not reduced in *jmj13* *clf* relative to *clf*, and *FT* expression is not analysed. It is not clear what tissue is being sampled i.e. leaves or whole seedlings, what age, how grown etc. Panel A also includes *jmj13* *clf* *swn/+* mutants, where suppression does not occur, but this is not referred to or explained in the text. Some analysis of H3K27me3 at targets could also be helpful.

We thank the reviewer for the constructive comments. The samples used in the target gene (*AP1*, *AP3*, *PI* etc.) analyses were ten-day-old seedlings grown

under 22°C LD conditions. We have added this information in the figure legend (Supp Fig 6h).

We also analyzed *FT* expression and found that *FT* expression level was reduced in *jmj13 clf* double mutant compared to *clf* single mutant. We added this data in the new Supp Fig. 6h and changed the legend accordingly.

The expression of *SEP3* and *AG* is not reduced in *jmj13 clf* relative to *clf* but the leaf curling and early flowering have a weak suppression in the *jmj13 clf* double mutant. This could be due to other H3K27me3 target genes causing these phenotypes, or due to REF6, ELF6, or functional redundancy of other H3K27me3 demethylases. We also added a short discussion about this point.

The suppression does not occur in *jmj13 clf swm/+* mutants. Again, this could be due to the functional redundancy with other H3K27me3 demethylases. Since this piece of data does not help much in explaining JMJ13 function, we deleted *jmj13 clf swm/+* from Supp Fig. 5d (now Supp Fig. 6).

Line 238 – 7.0 kb rather than 7.0 Mb intended here?

Thank you for the careful reading - it should indeed be 7.0 kb. We have corrected it.

Line 242 refers to H293 and E295 as iron binding, yet the structural figures (Fig 3 C) shows these residues binding Nickel nor Iron ions which is slightly confusing. Perhaps either Ni²⁺ or Fe²⁺ will do but clarification is needed.

Under physiological condition, the real metal ion required for jumonji family enzymes is Fe²⁺. So we state “H293 and E295 as iron binding” here. In our structural studies, we used the His-tag plus Nickel-column purification system for the high yield production of the protein. During the purification, the Ni²⁺ ion can replace the Fe²⁺ ion, which has been already shown by previous studies on the human histone demethylase KDM2A (Genes Dev 2014, 28:1758), KDM6A (Genes Dev 2011, 25:2266), and so on. So, we used Fe²⁺ to present the functional role of the metal ion and used Ni²⁺ to describe what we have observed in our structure. As suggested by the reviewer, we have added a clarification in our structure description to explain why we use Ni²⁺ to replace Fe²⁺ in the structure.

Line 260 – Authors suggest that FLM and SVP act downstream of JMJ13 based on experiments over expressing e.g. SVP transgene in *jmj13* mutant background and overcoming the early flowering. This implies they act in the same pathway, but I think other explanations are possible, e.g SVP acts in parallel on a common

target. No analysis of SVP or FLM expression in *jmj13* mutants is provided, no ChIP either of H3K27me3 levels or JMJ13 binding so it seems incomplete.

Line 275 – again, more is needed to support that *jmj13* acts via CO and GI, the genetics is consistent with this but on its own not compelling, expression analysis etc would help.

We thank the reviewer for the constructive comments and will response the two questions above together.

We agree that in addition to SVP and JMJ13 acting in the same pathway, SVP can also act in parallel on a common target. We added this point in the main text.

In the revised version, we performed new RNA-seq analysis and provided a list of differentially expressed genes. The RNA-seq showed that the expression of *LHY*, *CCA1* genes was up-regulated in *jmj13* (Supplementary data 1). This also suggests that many genes are regulated directly or indirectly by additional mechanisms, such as circadian rhythm. We added this information in the main text.

The genome browser data and real-time PCR results demonstrated that expression of *SVP* was decreased and *CO* was increased in *jmj13* (Fig. 8a and b).

We have also tried (ChIP-Seq) analysis but did not observe the difference between WT vs. *jmj13*. We added these negative data and the corresponding discussion in the Discussion section as follow.

“H3K27me3 levels in Col and *jmj13* by chromatin immunoprecipitation followed by sequencing (ChIP-Seq) were also analyzed, however no significant changes were observed (Supplementary Fig. 9), which is probably due to the JMJ13 does not possess direct DNA-binding domain and the low expression level of JMJ13. The H3K27me3 over-accumulation in *jmj13* was not visible (**Supplementary Fig. 5f**); on the other hand the H3K27me3 changes in *jmj13* are mild and no significant changes were detected in *FLM* and *SVP*, nor in *CO* and *GI*. All these are likely due to the functional redundancy of ELF6, REF6, and other H3K27me3 demethylases. Alternatively, the changes may occur specifically in a few genes, or occur predominantly in certain cell types, making it difficult to detect H3K27me3 changes in *jmj13* mutants. ”

See the following genome browser views of H3K27me3 signal.

Genome browser view of H3K27me3 signal for different genotypes at the FLM, SVP, CO and GI loci. Gene models from TAIR10 are shown at the bottom.

Line 280 – I think this should be KDM6 not KDM4?

In plants, the H3K27me3 demethylases REF6, ELF6, and JM13 are all classified into the KDM4 subfamily based on their JmjC domain sequence. Plants do not have known homologs of KDM6. To prevent confusion, we added a new supplementary table to compare the classification and the corresponding targets of the plant and animal demethylases. Please see the new Supplementary Table 1.

Line 288 – This part and Figure 7 should be presented in results rather than first mentioned in discussion.

As suggested, we have moved the section related to Fig. 7 in the results and rearranged the paper according to the reviewer's suggestion.

Reviewer #4 (Remarks to the Author):

In this work, Zheng et al. determined in the one hand the structure of the JM13 H3K27me3 demethylase catalytic domain-peptide complex, and in parallel, they also investigated the role of JM13 in flowering time control. They demonstrated that JM13 possesses H3K27me3 site-specific demethylase activity both in vitro and in vivo, as it was reported previously for this protein (Yan et al., 2018) and for two other H3K27me3 demethylases belonging to the same KDM4 subfamily, ELF6 and REF6, what diminish the novelty of these results. They further determined the crystal structures of JM13 in both H3K27me3 peptide-free and bound forms. They concluded that JM13 possesses a unique C4HCHC-type zinc finger, and not the previously predicted C5HC2-type zinc

finger domain, despite the two zinc finger types share a similar folding topology. Interestingly, they found that the overall structure of JMJ13 catalytic domain resembles the previously reported structures of human KDM5A/B/C and Arabidopsis JMJ14. They also provided detailed structural insight into the substrate specificity of JMJ13 showing that the substrate H3K27me3 peptide is specifically recognized by hydrogen bonding and hydrophobic stacking interactions. Besides, and to dissect the catalytic mechanism, they performed a thorough structure-based mutagenesis study, determining key residues for the activity of JMJ13. Nevertheless, further functional analyses in planta will be required to unravel how this putative plant KDM4 demethylase may use the KDM5 subfamily like-domain architecture to carry out the molecular functions of animal KDM6 demethylases.

To investigate the biological function of JMJ13, they produced several Arabidopsis transgenic lines overexpressing JMJ13, which show a strong global reduction in H3K27me3 levels and display pleiotropic defects, including acceleration of flowering time; in parallel, they analyzed Arabidopsis loss of function *jmj13* mutants, which also flower earlier, but do not show clear changes in the global H3K27me3 levels, possibly due to the redundancy with other H3K27me3 demethylases. That JMJ13 plays a role in flowering time control was already reported in an independent study by Yan et al., 2018, what again compromise at certain degree the novelty of some of the data presented in this manuscript.

Interestingly, Zheng et al. reveal that JMJ13 modulates flowering time in a temperature and day-length-dependent manner. They showed that impaired JMJ13 function leads to early flowering in both LD and SD conditions under warm temperature, but not in SD conditions with low temperature. They also provided genetic evidences suggesting that JMJ13 is a flowering repressor, acting upstream of both GI-CO and SVP-FLM pathways. Besides, the genetic interactions observed between the H3K27me3 methyltransferase CLF and JMJ13 demonstrate that the latter functions as an H3K27me3 demethylase that antagonizes CLF in vivo. However, the main drawback to this part of the paper is that there is not very much mechanistic insight for the putative role of JMJ13 in the temperature- and photoperiod dependent flowering regulation. I consider that these observations present an exciting start to answering the role of JMJ13 as an epigenetic repressor of flowering time, but clearly more work is needed to unveil the molecular mechanisms mediated by JMJ13 in the ambient temperature and photoperiod-dependent control of this developmental transition.

We thank the reviewer for the positive comments.

Major points:

1. What is the pattern of expression for JMJ13? (at tissue and temporal levels or under different environmental conditions of temperature and photoperiod?). There is already some information published in a previous report (Yan et al., 2018). Once established these patterns, the profile of specific gene regulation by JMJ13 under particular growing conditions of temperature and photoperiod should be explored by following RNA-seq approaches with the *jmj13* mutant.

We thank the reviewer for the constructive comments. We used 7-day-old transgenic seedlings grown under 22°C LD, 22°C SD and 28°C SD to test the expression level of JMJ13. We observed clearly that the JMJ13-GUS was enhanced under long-day or higher temperature and GUS staining data were added into Supp Fig. 8. JMJ13 is mainly expressed in apical meristem including newly grown euphylla and root tip tissues.

In addition, we analyzed the mRNA (Fig. 8c) and protein (Fig. 8d) levels of JMJ13 under different environmental conditions as suggested. Due to the low level of JMJ13, here, we used 7-day old transgenic seedlings (*JMJ13ox-13*) grown under 22°C SD, 22°C LD, and 28°C SD to test the protein accumulation of JMJ13. We found that long-day conditions and higher temperature induced the expression of *JMJ13* and increased accumulation of JMJ13 (Fig. 8 c and d). RNA-seq was performed for *Col* and *jmj13* grown under 22°C LD conditions (Fig. 8a and Supplementary data 1). These results indicated that JMJ13 plays an important role in temperature- and photoperiod-mediated flowering time regulation pathways, which is consistent with our genetic data. We added these data to our revised manuscript.

2. Which are the direct targets of JMJ13 in flowering time control? The identification of genome wide binding sites of JMJ13 by ChIP-seq approaches involving the HA-tagged JMJ13 complemented plants shown in Fig.4, will be highly recommended to unveil putative target genes of this protein (or at least they should demonstrate by ChIP-PCR the direct binding of JMJ13 to a number of putative targets related to flowering time control that are misregulated in the *jmj13* mutant.

The direct targets of JMJ13 under three conditions were detected by Chromatin Immunoprecipitation Sequence (ChIP-Seq). Unfortunately, there was no noticeable binding signal probably due to the low expression of JMJ13, or the binding may be specific to a few genes, or in certain cell types. The direct targets of JMJ13 will be an important issue to explain the regulation mechanism and we will focus on it in our future work.

3. It would be also highly informative to provide insight on the mechanisms of JMJ13 target site recruitment given that apparently JMJ13 has no DNA-binding domain.

Currently, there are two major targeting mechanisms, as shown by JMJ14 through NAC family transcription factors or by JMJ12/REF6 through its ZnF domains to bind specific DNA motifs. But unfortunately, we did not succeed in identify the potential targeting mechanism for JMJ13, as we did not find any JMJ13 binding protein and JMJ13 does not possess DNA-binding domain. This is in consistent with our failure of ChIP-seq experiment. We think the low expression level of JMJ13 and absence of DNA binding domain may induce some technical problem for us to study the direct targeting and chromatin association mechanism for JMJ13. This issue is one of our topics for future studies.

4. Temporal dynamics of JMJ13 –mediated H3K27 demethylation at genome wide level (or at least in a number of putative targets related to flowering time control) should be analyzed under different conditions of temperature and photoperiod in the *jmj13* mutant.

We thank the reviewer for the constructive comments. The H3K27me3 accumulation in *Col* and *jmj13* under three conditions were detected by Chromatin Immunoprecipitation Sequence (ChIP-Seq). However, there was no noticeable changing in H3K27me3 accumulation in *FLM*, *SVP* and *CO*, *GI*. This may due to functional redundancy of ELF6, REF6 and other H3K27me3 demethylases, as the H3K27me3 changes in *jmj13* are mild. Also, the changes may be specific to a few genes, or occurred predominantly in certain cell types, making it difficult to detect H3K27me3 changes in *jmj13* mutants. We added this in the discussion.

5. 5-1 The *elf6* and the *jmj13* loss-of-function mutants displayed early flowering phenotypes while *ref6* mutants flower later than WT, suggesting that these H3K27me3 demethylases can influence flowering via different pathways. However, it would be interesting to check the possible contribution of ELF6 and REF6 in the temperature- and photoperiod dependent flowering control by analyzing the flowering behaviour of *elf6*, *ref6* or double and triple mutant combinations.

We thank the reviewer for the constructive comments. Unlike *jmj13*, the plants with mutations of ELF6 (*jmj11*) showed early and late flowering phenotypes, respectively, regardless to different temperatures and day length (Supp Fig. 7). Furthermore, we obtained the double mutants and the triple mutant. The

flowering time was analyzed under 22°C LD conditions. The *ref6* mutation could suppress the early flowering in *jmj13* and *elf6* (*jmj11*) single and double mutants (shown in the following figure). The results showed that REF6 plays a dominant function in flowering time control, as described in Yan et al, 2018.

5-2 Also it may be worth to assess the flowering behavior of JMJ13ox and *clf* mutants under the same experimental conditions.

JMJ13ox plants displayed global H3K27me3 decreased as shown in Supp Fig. 5f. This drastic H3K27me3 decrease caused abnormal expression of many H3K27me3 target genes, which may not be the direct targets of JMJ13.

Using *clf* mutants to assess flowering behavior is a good idea to test whether H3K27me3 abundance affects the temperature- and photoperiod-mediated flowering time. But early flowering is so severe in *clf*, we might not be able to observe the differences between different growth conditions.

6. JMJ13ox plants displayed dwarf seedlings, early flowering and upward curling leaves pleiotropic phenotypes. Loss of function *jmj13* mutants are also early flowering. The authors should discuss this observation, where the loss- and the gain-of-function of JMJ13 cause the same flowering phenotype. Besides, the genetic interaction observed for JMJ13 and CLF in the control of flowering time, where the double mutant *jmj13 clf* flowers later than any of the *jmj13* or *clf* single parentals, that are both early flowering mutants, and similarly to WT plants (Suppl. Figure 5d), should be explained in more detail. Same for the genetic interaction revealed for *jmj13clf swn +/-* plants regarding flowering time that it is shown in Suppl. Figure 5d and that it is not even described in the manuscript.

JMJ13ox plants displayed a global reduction in H3K27me3 as shown in Supp Fig. 5f. The H3K27me3 reduction caused ectopic expression of H3K27me3 target genes, which may not be the direct targets of JMJ13. As JMJ13 has both direct and indirect targets and we were not able to identify the direct target for JMJ13 by ChIP-seq, we cannot figure out the downstream signal network for JMJ13 and the events happened in JMJ13ox and *jmj13*. Therefore, we cannot get a clear answer for this. We added some discussion about this in the main text.

We think that the double mutant *jmj13 clf* flowering later than any of the *jmj13* or *clf* single mutant is probably due to the complex regulation system and the double can be considered as weaker version for the single. We have added a short discussion about this.

The suppression does not occur in *jmj13 clf swn/+* mutants. Again, this could be due to the functional redundancy with other H3K27me3 demethylases. The results of *jmj13 clf swn/+* plants in Supp Fig. 5d (now Supp Fig. 6g) does not help much in explaining JMJ13 function, so we deleted *jmj13 clf swn/+* from Supp Fig. 5d (now Supp Fig. 6g).

Minor points:

1. Indicate the *svp* allele used in the studies performed in Fig.5.

The *svp* allele used in our study is *svp-32* (SALK_072930). We indicated this in the text and the Materials and Methods.

2. It would be highly informative to include FT and FLC expression data in the analysis performed in Suppl. Fig. 5 e, involving *jmj13* and combinations with *clf* mutation.

We added *FT* expression in Supp Fig. 5e (now Supp Fig. 6h) and changed the title accordingly. In our former RNA-seq data (shown in the following Figure screenshot from a genome browser), *JMJ13* has no effect on *FLC* sites. So, *FLC* was not included in the real-time PCR analysis in Suppl. Fig. 5 e (now Supp Fig 6h).

3. Writing throughout the manuscript should be revised. A number of suggestions are provided below, but additional editing is strongly advised.

Thank you, we have had the paper edited by a native English speaker and professional editor.

Lane 51 long-day should be "LD"

We have corrected it to 'LD' in line 51 and following lines, as well as 'SD'.

Lane 68-69. Please rephrase to: ... in a temperature –dependent manner, and a strong binding of PIF4 to FT depends on the eviction of H2A.Z nucleosomes

We have rephrased here as suggested.

Lane 123 MOLDI I should be changed by "MALDI"

We have corrected it.

Lane 141 Add "was determined" after resolution.

We have added it.

Lane 161 Change "locating" by "located".

We have corrected it.

Lane 230 Change "antagonist" by " antagonize".

We have corrected it.

Lane 230 Change "methyltransferases" by " methyltransferase ".

We have corrected it.

Lane 252. Please rephrase to: JMJ13 negatively modulates flowering time under LD and in SD under high temperatures.

We have rephrased it as suggested.

Lane 360 Change "Colombia " by "Columbia "

We have changed it.

Lane 634-635: Rephrased to: "key residues involved in peptide binding and catalysis is decreasing the activity....."

We have rephrased it as suggested.

Lane 680: Rephrased to: Fig 6. The role of JMJ13 in photoperiod –regulated flowering...

We have rephrased it as suggested.

Several Col all over the text are in italics and should not be.

We have corrected them.

Suppl Fig 4 panel f: *jmj* mutation should be written in lower case and italics

We have corrected it.

Suppl Fig 5 panel d: include *jmj13clfswn*+/- in the figure legend and describe the result observed for the flowering time of this plant in the manuscript text.

This does not help much in explaining JMJ13 functional mechanisms so we deleted *jmj13 clf swn*+ in Supp Fig. 5d (now Supp Fig. 6g).

REVIEWERS' COMMENTS:

Reviewer #3 (Remarks to the Author):

The authors have added some extra data and generally improved the m/s in response to comments. Overall my impression remains similar to before, i.e. that the structural part is interesting and strong, the biochemistry and functional methylation assays are also well done. The part on flowering time is not greatly resolved by the additional data and remains very unclear mechanistically. I agree that the data supports a minor role for JMJ13 in flowering time control, but do not feel the case is strong that this operates via CO/GI/FLM and SVP particularly given that no changes in H3K27me3 are shown in *jmj13* lof or direct binding of JM13 is shown to targets. The genetics experiments in my view do not resolve whether JMJ13 acts via these genes, or simply in parallel on e.g. FT. The authors have done experiments such as H3K27me3 ChIP seq, but these have not been sensitive enough to resolve effects of JMJ13 loss of function. Personally I don't think this should necessarily prevent publication, and clearly the authors have tried hard to address these issues, but I think they should be more circumspect in conclusions and more consistent. For example, the authors state that they have added a statement in main text that JMJ13 may act in parallel to FLM and SVP but this applies to the results section, the discussion (eg line 377) and abstract persist in stating that the results show that JMJ13 acts upstream of FLM and SVP.

There are some oddities introduced in the revised version. For example, the discussion of the structural data is now in the results section rather than in the discussion, which focuses on the mechanism of flowering time regulation. ChIP seq results are presented in discussion but not in results (line 383 - 393). Lin 272 - 274 and 286 - 289 are not easy to follow and may need revision

Reviewer #4 (Remarks to the Author):

In this update version, the authors have tried to improve the manuscript in terms of supporting the role of JUMONJI 13 as a temperature and photoperiod dependent epigenetic flowering repressor, addressing some of the major concerns raised by this reviewer, such as the pattern of expression of the JMJ13 gene under different conditions of temperature and photoperiod, the profile of specific gene regulation by JMJ13 or the temporal dynamics of JMJ13-mediated H3K27 demethylation at global level.

The new information provided reveals that LD photoperiods and higher temperatures induce the expression of JMJ13 and the accumulation of JMJ13 protein. Besides, new RNA-seq data involving *jmj13* grown under 22°C LD conditions versus the corresponding wild type plants were included in this updated version. These results support a role for JMJ13 in temperature- and photoperiod-mediated flowering time regulation pathways, which is consistent with the genetic data previously provided.

When they performed ChIP-seq to determine the accumulation of H3K27me3 levels in Col and *jmj13* under three different growing conditions, the obtained data showed no noticeable changes in H3K27me3 accumulation in FLM, SVP, CO and GI chromatin in the *jmj13* mutant, although they report lower levels of expression for SVP and higher levels for CO in *jmj13*, that in principle, cannot be explained by altered H3K27me3 levels. They conclude that functional redundancy between JMJ13 and ELF6, REF6 and other H3K27me3 demethylases may exist, making it difficult to detect clear H3K27me3 changes in *jmj13* mutants. In accordance with this, Yan et al. 2018 (Nature Plants) already reported that only global H3K27me3 levels were strongly elevated in the triple *elf6ref6jmj13* mutant compared to wild-type, single and double mutants. Discuss this point in the final version. Zheng et al also propose that the changes in H3K27me3 due to JMJ13 may be specific to a few genes, or occurred predominantly in certain cell types. But did they find any genes with conspicuous changes in H3K27me3 in *jmj13* mutant in their ChIP analysis? And were they misregulated in *jmj13*?

They also tried to address the issue concerning the identification of direct targets of JMJ13, but

they did get no noticeable binding signal in the ChIP-seq performed due to technical limitations, either by low expression of JMJ13, or that the JMJ13 binding may be specific to a reduced number of genes or only occurs at particular cell types, making it difficult to unravel putative targets with this approach. They either did not succeed in identifying the potential targeting mechanism for JMJ13, as they did not find any JMJ13 binding protein. I consider that the identification of the JMJ13 targets or partners is an essential issue that may help to explain the regulatory mechanisms exerted by JMJ13. Unfortunately, the authors were not able to provide any experimental evidence related to these issues.

I also suggested to check the possible contribution of ELF6 and REF6 in the temperature- and photoperiod-dependent flowering control by analyzing the flowering behaviour of *elf6*, *ref6* or double and triple mutant combinations. In the "Response to reviewers" letter the authors provided information regarding the flowering time for the mutant combinations grown only under LD 22°C, showing that at this condition, *elf6* and *jmj13* are partially redundant for flowering time and that *ref6* mutations suppress the early flowering phenotype of *elf6* and *jmj13*. Maybe it is worth considering to include this information in the final version of the manuscript and discuss it together with the data provided in Supp Fig 7 for *elf6* and *jmj13* mutants (although data with *ref6* is missing) grown under different temperature and photoperiodic conditions.

Finally, to the suggestion of assessing the flowering behaviour of *clf* mutants under different temperature and photoperiod conditions, the authors reply that it could be a good idea to test whether H3K27me3 abundance affects the temperature- and photoperiod-mediated flowering time. However, they state that the early flowering phenotype is so severe in *clf*, that it might prevent to observe differences between distinct growth conditions: I do not agree on this comment because the flowering time of *clf* is almost identical to the *jmj13* mutant as they showed in Supp Fig 6 g, and they were able to provide reliable flowering data concerning *jmj13*.

Minor points

1. Reword the phrase in line 273-274.
2. Line 282 remove AGAMOUS. Leave only the acronym AG
3. Line 324 change *flm-3* by *flm* (also in Fig. 6). It is the only mutant allele for which the number of the allele is indicated along the manuscript.
4. Line 409. Indicate the *lhp1* allele used in the study.
5. Line 844: change "was analysis" to: was analysed
6. Change the lettering in Y-axis of Fig 7 e to: FT relative expression
7. Lines 860 and 862 Include "levels" after the transcriptional.
8. Change the lettering in Y-axis of Fig 8b to: relative transcript expression
9. Supp Fig 5 legend. There is a mistake. Weak to strong phenotype are shown from right to left, not to left to right.

We thank all the reviewers for their positive comments and constructive suggestions. We have revised our manuscript according to the suggestions. A point-to-point response is below with our responses highlighted in red.

REVIEWERS' COMMENTS:

Reviewer #3 (Remarks to the Author):

The authors have added some extra data and generally improved the m/s in response to comments. Overall my impression remains similar to before, i.e. that the structural part is interesting and strong, the biochemistry and functional methylation assays are also well done. The part on flowering time is not greatly resolved by the additional data and remains very unclear mechanistically. I agree that the data supports a minor role for JMJ13 in flowering time control, but do not feel the case is strong that this operates via CO/GI/FLM and SVP particularly given that no changes in H3K27me3 are shown in *jmj13* or direct binding of JM13 is shown to targets. The genetics experiments in my view do not resolve whether JMJ13 acts via these genes, or simply in parallel on e.g. FT. The authors have done experiments such as H3K27me3 ChIP seq, but these have not been sensitive enough to resolve effects of JMJ13 loss of function. Personally I don't think this should necessarily prevent publication, and clearly the authors have tried hard to address these issues, but I think they should be more circumspect in conclusions and more consistent. For example, the authors state that they have added a statement in main text that JMJ13 may act in parallel to FLM and SVP but this applies to the results section, the discussion (eg line 377) and abstract persist in stating that the results show that JMJ13 acts upstream of FLM and SVP.

We thank reviewer for the constructive comments. We agree with the reviewer that our data are only consistent with JMJ13 acts upstream or in parallel to SVP/FLM and/or CO/GI, but we cannot get this conclusion just based on the current data. Additional experiments are needed to figure out the detail mechanism. Therefore, we weakened our statement in the revised manuscript by replacing the definitive statement by the possibility and simply describing the results without further interpretation. In addition, we kept Fig 1-5 in the main text as they are solid data to characterize JMJ13. To prevent misinterpretation, the original Fig 6-8 were moved to the Supplementary data as support data to indicate the possibility that JMJ13 is possible to act upstream or in parallel FLM/SVP and GI/CO.

There are some oddities introduced in the revised version. For example, the discussion of the structural data is now in the results section rather than in the discussion, which focuses on the mechanism of flowering time regulation. ChIP seq results are presented in discussion but not in results (line 383 - 393). Lin 272 - 274 and 286 - 289 are not easy to follow and may need revision

The structure comparison between JMJ13 and other histone demethylase is aim to further describe the key molecular feature of JMJ13. As suggested by another reviewer, we put this section in the Results section. In this arrangement, the structure related sections are focused together which is easier for the reader to follow. Or else, the manuscript looks incoherent by switching between structure and function. To prevent further misleading, we have moved the ChiP-seq data in the results section as suggested by the reviewer to only leave the general discussions but not details in the Discussion section.

The sentence 272-274 appears irrespectively here and was removed from the final version. The sentence 286 - 289 is a simple discussion of the sentence above and looks like repeat discussion. So, we delete this sentence, too.

Reviewer #4 (Remarks to the Author):

In this update version, the authors have tried to improve the manuscript in terms of supporting the role of JUMONJI 13 as a temperature and photoperiod dependent epigenetic flowering repressor, addressing some of the major concerns raised by this reviewer, such as the pattern of expression of the JMJ13 gene under different conditions of temperature and photoperiod, the profile of specific gene regulation by JMJ13 or the temporal dynamics of JMJ13-mediated H3K27 demethylation at global level. The new information provided reveals that LD photoperiods and higher temperatures induce the expression of JMJ13 and the accumulation of JMJ13 protein. Besides, new RNA-seq data involving *jmj13* grown under 22°C LD conditions versus the corresponding wild type plants were included in this updated version. These results support a role for JMJ13 in temperature- and photoperiod-mediated flowering time regulation pathways, which is consistent with the genetic data previously provided. When they performed Chip-seq to determine the accumulation of H3K27me3 levels in *Col* and *jmj13* under three different growing conditions, the obtained data showed no noticeable changes in H3K27me3 accumulation in *FLM*, *SVP*, *CO* and *GI* chromatin in the *jmj13* mutant, although they report lower levels of expression for *SVP* and higher levels for *CO* in *jmj13*, that in principle, cannot be explained by altered H3K27me3 levels. They conclude that functional redundancy between JMJ13 and *ELF6*, *REF6* and other H3K27me3 demethylases may exist, making it difficult to detect clear H3K27me3 changes in *jmj13* mutants. In accordance with this, Yan et al. 2018 (Nature Plants) already reported that only global H3K27me3 levels were strongly elevated in the triple *elf6 ref6 jmj13* mutant compared to wild-type, single and double mutants. Discuss this point in the final version.

We thank reviewer for the constructive comments. As suggested, we have added “These observations are in consistent with recent report that only global H3K27me3 levels were strongly elevated only in the triple *elf6 ref6 jmj13* mutant in *Arabidopsis*³⁸.” (Yan et al., Nature Plants, 2018).

Zheng et al also propose that the changes in H3K27me3 due to JMJ13 may be specific to a few genes, or occurred predominantly in certain cell types. But did they find any genes with conspicuous changes in H3K27me3 in *jmj13* mutant in their CHIP analysis? And were they misregulated in *jmj13*?

We have not been able to find cell type specific genes which are misregulated in *jmj13*. We thank the reviewer to point out such indefinite speculation. To avoid misleading, we remove “Alternatively, the changes may occur specifically in a few genes, or occur predominantly in certain cell types, making it difficult to detect H3K27me3 changes in *jmj13* mutants”. Data from Yan et al (Nature Plants, 2018) also support our observation. Therefore, we add “Global H3K27me3 levels were only increased in the triple *elf6 ref6 jmj13* mutant, conforming that JMJ13, REF6, and ELF6 are functional redundant. ..., which partial explained that there is no over-accumulation of H3K27me3 in *jmj13*.” in the Discussion section.

They also tried to address the issue concerning the identification of direct targets of JMJ13, but they did get no noticeable binding signal in the CHIP-seq performed due to technical limitations, either by low expression of JMJ13, or that the JMJ13 binding may be specific to a reduced number genes or only occurs at particular cell types, making difficult to unravel putative targets with this approach. They either did not succeed in identifying the potential targeting mechanism for JMJ13, as they did not find any JMJ13 binding protein. I consider that the identification of the JMJ13 targets or partners is an essential issue that may help to explain the regulatory mechanisms exerted by JMJ13. Unfortunately, the authors were not able to provide any experimental evidence related to these issues.

We totally agreed with the reviewer that identification of the JMJ13 targets is an essential issue to fully understand where JMJ13 acts globally. Here in this paper, we mainly focus on how H3K27me3 was recognized and removed by JMJ13 in plants. We would like to leave the targeting issue for further investigation. Therefore, we have made a comment as “Further work would be needed to determine how JMJ13 affects flowering time, which JMJ13 targets that might be phenotypically relevant, and ...” in the Discussion section.

I also suggested to check the possible contribution of ELF6 and REF6 in the temperature- and photoperiod dependent flowering control by analyzing the flowering behaviour of *elf6*, *ref6* or double and triple mutant combinations. In the “Response to reviewers” letter the authors provided information regarding the flowering time for the mutant combinations grown only under LD 22°C, showing that at this condition, *elf6* and *jmj13* are partially redundant for flowering time and that *ref6* mutations suppress the early flowering phenotype of *elf6* and *jmj13*. Maybe it is worth considering to include this information in the final version of the manuscript and discuss it together with the data

provided in Supp Fig 7 for *elf6* and *jmj13* mutants (although data with *ref6* is missing) grown under different temperature and photoperiodic conditions.

We agreed with the reviewer suggest. In this paper, we want to use *JMJ13* as an example to demonstrate on how H3K27me3 was recognized and removed in plants. As shown by Yan et al., (Nature Plants, 2018) that *REF6* plays a major role in determining the distribution of H3K27me3, and *JMJ13* plays a minor role in flowering time regulation. Since every H3K27 demethylase acts on thousands of genes and flowering-time genes are numerous, which could be positively or negatively regulate flowering time. Therefore, by scoring the double mutants may not greatly help us far enough to understand the detailed mechanism of *JMJ13*. This issue would be resolved in our future work to analyze the differences and similarities function between *ELF6*, *REF6* and *JMJ13*. We would like to refer this as our future work like “Further work would be needed to determine ..., and what are the differences and similarities among *ELF6*, *REF6*, and *JMJ13*” in the Discussion section.

Finally, to the suggestion of assessing the flowering behaviour of *clf* mutants under different temperature and photoperiod conditions, the authors reply that it could be a good idea to test whether H3K27me3 abundance affects the temperature- and photoperiod-mediated flowering time. However, they state that the early flowering phenotype is so severe in *clf*, that it might prevent to observe differences between distinct growth conditions: I do not agree on this comment because the flowering time of *clf* is almost identical to the *jmj13* mutant as they showed in Supp Fig 6 g, and they were able to provide reliable flowering data concerning *jmj13*.

clf is a strong mutant and acts on thousands of genes and many of those could directly or indirectly regulate flowering time. Therefore, by scoring the flowering time of *clf jmj13* double mutants will probably be hard to illustrate the mechanism of *JMJ13*. We would like to perform a detail analysis of the crosstalk between *CLF* and *JMJ13* in our future work.

Minor points

1. Reword the phrase in line 273-274.

The sentence appears irrespectively here and was removed from the final version.

2. Line 282 remove *AGAMOUS*. Leave only the acronym *AG*

Corrected.

3. Line 324 change *flm-3* by *flm* (also in Fig. 6). It is the only mutant allele for which the number of the allele is indicated along the manuscript.

Corrected.

4. Line 409. Indicate the *lhp1* allele used in the study.

The *lhp1* mutant used in this study was *tf12-1* as describe by Larsson *et al* (1998, Genetics).

5. Line 844: change " was analysis" to: was analysed

Corrected.

6. Change the lettering in Y-axis of Fig 7 e to: FT relative expression

Updated as suggested.

7 Lines 860 and 862 Include "levels" after the transcriptional.

Corrected.

8. Change the lettering in Y-axis of Fig 8b to: relative transcript expression

Updated as suggested.

9. Supp Fig 5 legend. There is a mistake. Weak to strong phenotype are shown from right to left, not to left to right.

Corrected.